# CLEANBA: A REPRODUCIBLE AND EFFICIENT DISTRIBUTED REINFORCEMENT LEARNING PLATFORM

**Shengyi Huang**[‡]🤗  **Jiayi Weng**[*]  **Rujikorn Charakorn**[♯]  **Min Lin**[△]
**Zhongwen Xu**[◇]  **Santiago Ontañón**[‡§]

[‡]Drexel University  🤗Hugging Face  [§]Google  [♯]VISTEC  [△]Sea AI Lab  [◇]Tencent AI Lab
`costa.huang@outlook.com`

## ABSTRACT

Distributed Deep Reinforcement Learning (DRL) aims to train autonomous agents in less wall-clock time by leveraging more computational resources. Despite recent progress in the field, reproducibility issues have not been sufficiently explored. This paper first shows that the typical actor-learner framework can have reproducibility issues even if hyperparameters are controlled. We then introduce Cleanba, a new open-source platform for distributed DRL that proposes a highly reproducible architecture. Cleanba implements highly optimized distributed variants of PPO (Schulman et al., 2017) and IMPALA (Espeholt et al., 2018). Our Atari experiments show that these variants can obtain equivalent or higher scores than strong IMPALA baselines in `moolib` and `torchbeast` and PPO baseline in CleanRL. However, Cleanba variants present 1) shorter training time and 2) more reproducible learning curves in different hardware settings. Cleanba's source code is available at `https://github.com/vwxyzjn/cleanba`.

## 1 INTRODUCTION

Deep Reinforcement Learning (DRL) is a technique to train autonomous agents to perform tasks. In recent years, it has demonstrated remarkable success across various domains, including video games (Mnih et al., 2015), robotics control (Schulman et al., 2017), chip design (Mirhoseini et al., 2021), and large language model tuning (Ouyang et al., 2022). Distributed DRL (Espeholt et al., 2018; 2020) has also become a fast-growing paradigm that trains agents in less wall-clock time by leveraging more computing resources. Despite recent progress, reproducibility issues in distributed DRL have not been sufficiently explored. This paper introduces Cleanba, a new platform for distributed DRL that addresses reproducibility issues under different hardware settings.

Reproducibility in DRL is a challenging issue. Not only are DRL algorithms brittle to hyperparameters and neural network architectures (Henderson et al., 2018), implementation details are often crucial for successfully applying DRL but frequently omitted from publications (Engstrom et al., 2020; Andrychowicz et al., 2021; Huang et al., 2022a). Reproducibility issues in distributed DRL are under-studied and arguably even more challenging. In particular, most high-profile distributed DRL works, such as Apex-DQN (Horgan et al., 2018), IMPALA (Espeholt et al., 2018), R2D2 (Kapturowski et al., 2019), and Podracer Sebulba (Hessel et al., 2021) are not (fully) open-source. Furthermore, earlier work pointed out that more actor threads not only improve training speed but cause reproducibility issues – different hardware settings could impact the data efficiency in a non-linear fashion (Mnih et al., 2016).

In this paper, we present a more principled approach to distributed DRL, in which different hardware settings could make training speed slower or faster but do not impact data efficiency, thus making scaling results more reproducible and predictable. We first analyze the typical actor-learner architecture in IMPALA (Espeholt et al., 2018) and show that its parallelism paradigm could introduce reproducibility issues due to the concurrent scheduling of different actor threads. We then propose a more reproducible distributed architecture by better aligning the parallelized actor and learner's compu-

---

[*]Currently at OpenAI.

tations. Based on this architecture, we introduce our Cleanba (meaning **Clean**RL-style (Huang et al., 2022b) Podracer Sebul**ba**) distributed DRL platform, which aims to be an easy-to-understand distributed DRL infrastructure like CleanRL, but also be scalable as Podracer Sebulba. Cleanba implements a distributed variant of PPO (Schulman et al., 2017) and IMPALA (Espeholt et al., 2018) with JAX (Bradbury et al., 2018) and EnvPool (Weng et al., 2022). Next, we evaluate Cleanba's variants against strong IMPALA baselines in `moolib` (Mella et al., 2022) and `torchbeast` (Küttler et al., 2019) and PPO baseline in CleanRL (Huang et al., 2022b) on 57 Atari games (Bellemare et al., 2013). Here are the key results of Cleanba:

1. **Strong performance**: Cleanba's IMPALA and PPO achieve about 165% median human normalized score (HNS) in Atari with sticky actions, matching `monobeast` IMPALA's 165% median HNS and outperforming `moolib` IMPALA's 140% median HNS.

2. **Short training time**: Under the 1 GPU 10 CPU setting, Cleanba's IMPALA is **6.8x faster** than `monobeast`'s IMPALA and **1.2x faster** than `moolib`'s IMPALA. Under a max specification setting, Cleanba's IMPALA (8 GPU and 40 CPU) is **5x faster** than `monobeast`'s IMPALA (1 GPU and 80 CPU) and **2x faster** than `moolib`'s IMPALA (8 GPU and 80 CPU).

3. **Highly reproducible**: Cleanba shows predictable and reproducible learning curves across 1 and 8 GPU settings given the same set of hyperparameters, whereas `moolib`'s learning curves can be considerably different, even if hyperparameters are controlled to be the same.

4. **Highly scalable**: Cleanba can linearly scale to multi-node settings, allowing the researchers to leverage hundreds of GPUs (Appendix F).

To facilitate more transparency and reproducibility, we have made available our source code at https://github.com/vwxyzjn/cleanba.

## 2 BACKGROUND

**Distributed DRL Systems** Utilizing more computational power has been an attractive topic for researchers. Earlier DRL methods like DQN (Mnih et al., 2015) were synchronous and typically used a single simulation environment, which made them slow and inefficient in using hardware resources. A3C (Mnih et al., 2016) spawns multiple actor threads; each interacts with its own copy of the environment and asynchronously accumulates gradient. To make distributed DRL more scalable, IMPALA decouples the actors and the learners (Espeholt et al., 2018; 2020). The actors produce training data asynchronously, while the learners produce new agent parameters, which are transferred asynchronously to the actor. Actor-learner systems can achieve higher throughput and shorter training wall time than A3C. Additional distributed actor-learner systems include GA3C (Babaeizadeh et al., 2017), IMPALA (Espeholt et al., 2018), Apex-DQN (Horgan et al., 2018), R2D2 (Kapturowski et al., 2019), and Podracer Sebulba (Hessel et al., 2021).

**Reproducibility Issues with Different Hardware Settings** Empirical evidence suggests that increasing the number of actor threads can enhance the training speed in distributed DRL (Mnih et al. (2016, Fig. 4)). However, this augmentation is not without its complications. It also impacts data efficiency and final Atari scores (Mnih et al. (2016, Fig. 3)), and these effects could manifest in a non-linear manner. While the authors found the side effects of value-based asynchronous methods to be positive and improve data efficiency, the side effects of contemporary distributed DRL systems, such as IMPALA, Apex-DQN, and R2D2, across various hardware configurations, have not been sufficiently explored.

**Open-source Distributed DRL Infrastructure** While many distributed DRL algorithms are not open-source, there have been many notable distributed DRL replications in the open-source software (OSS) community. These efforts include SEED RL (Espeholt et al., 2020), `rlplyt` (Stooke & Abbeel, 2018), Decentralized Distributed PPO (Wijmans et al., 2020), Sample Factory (Petrenko et al., 2020), HTS-RL (Liu et al., 2020), `torchbeast` (Küttler et al., 2019), and `moolib` (Mella et al., 2022). Many of them have shown high throughput and good empirical performance in select domains. Nevertheless, most of them either do not have evaluations on 57 Atari games or have various hardware restrictions, leading to reproducibility concerns. `moolib` is the only OSS infras-

**IMPALA Actor-Learner Architecture**

```
1   batch_size = 32
2   agent = Agent()
3   rollout_Q = queue()
4
5   def actor():
6       while True:
7           data = rollout(agent.param, 1)
8
9
10          rollout_Q.put(data)
11  def learner():
12      for _ in range(1, ITER):
13          data = rollout_Q.get_many(batch_size)
14          agent.learn(data)
15          broadcast_to_actors(agent.param)
16  for _ in range(num_actors):
17      thread(actor).start()
18  thread(learner).start()
```

**Cleanba's architecture**

```
1   batch_size = 32
2   agent = Agent()
3   rollout_Q = queue(max_size=1)
4   param_Q = queue(max_size=1)
5   def actor():
6       for i in range(1, ITER):
7           if i != 2:
8               params = param_Q.get()
9           data = rollout(params, batch_size)
10          rollout_Q.put(data)
11  def learner():
12      for _ in range(1, ITER):
13          data = rollout_Q.get()
14          agent.learn(data)
15          param_Q.put(agent.param)
16  param_Q.put(agent.param)
17  thread(actor).start()
18  thread(learner).start()
```

Figure 1: The pseudocode for IMPALA architecture (left) and Cleanba's architecture (right). Colors are used to highlight the code differences between the two architectures. The `rollout(params, num_envs)` function collects rollout data on `num_envs` independent environments for `num_steps` steps.

tructure that has both evaluations on 57 Atari games in the standard 200M frames setting and can scale beyond a single GPU setting[1].

## 3 REPRODUCIBILITY ISSUES IN IMPALA

This section shows that IMPALA (Espeholt et al., 2018) has non-determinism by nature, which arises from the concurrent scheduling of different actor threads. This non-determinism could further cause subtle reproducibility issues.

A natural question arises: *what happens when the learner produces a new policy while the actor is in the middle of producing a trajectory?* It turns out multiple policy versions could contribute to the actor's rollout data in line 7 of the IMPALA architecture Figure 1. Typically, the faster the policy updates, the more frequently the policies are transferred. However, this impacts the rollout data construction in a non-trivial way. From a reproducibility point of view, it is important to realize the frequency at which the policies are updated is a source of non-determinism.

However, non-determinism can be desirable in parallel programming because they make programs faster without making outputs significantly different. For example, some of NVIDIA's CuDNN operations are inherently non-determinisitic[2]. What is more important is to investigate if this non-determinism could cause reproducibility issues in terms of learning curves. To this end, we manufacture a specific experiment that magnifies this non-determinism in `monobeast`'s IMPALA. For the control group, we

1. decreased the number of trajectories in the batch from 32 to 8 to reduce training time, thus making the actor's policy updates more frequent;

2. used 80 actor threads and increased `monobeast`'s default unroll length from 20 to 240 to increase the chance of observing the actor's policy updates in the middle of a trajectory.

---

[1]While SEED RL also has evaluations on 57 Atari games and scale beyond 1 GPU, SEED RL trained the agents for 40 billion frames 40 hours per game.

[2]https://docs.nvidia.com/deeplearning/cudnn/developer-guide/index.html#reproducibility

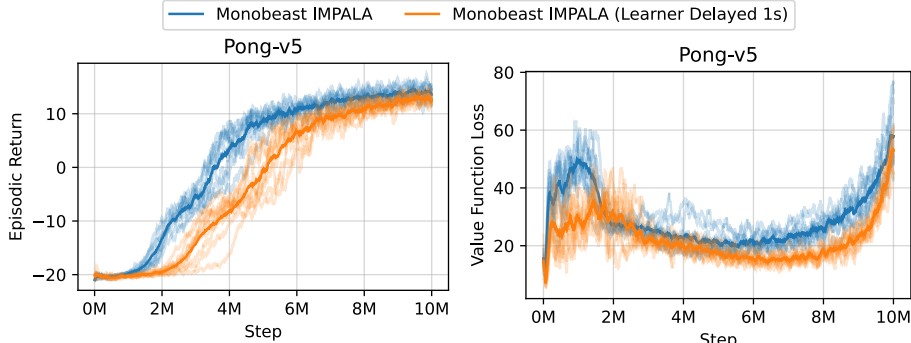

Figure 2: **IMPALA's reproducibility issue under different "speed" settings** — The y-axes show the episodic return and value function loss of two sets of `monobeast` experiments that use the *exact same hyperparameters*, but the orange set of experiments has its learner update manually delayed for 1 second to simulate slower learner updates. Note the learning curves across 10 random seeds are non-trivially different, implicating hyperparameters in IMPALA alone cannot always ensure good reproducibility.

For the experimental group, we used the above setting but *manually slowed down the policy broadcasting* by sleeping the learner for 1 second after the policy updates in order to simulate a case where the learner is significantly slower (such as when running the learner on CPU).

We found that in the control group, the actors, on average, changed their policy versions **12-13 times** in the middle of the 240-length trajectory. In the experimental group, because of the manual slowdown in broadcasting the learner's policy, the actors, on average, changed the policy **one time**. We note that the results vary on different hardware settings as well. For example, the control group changed their policy versions, on average, eight times when using 40 actor threads. We noted that in `moolib`, the actor's policy could also change mid-rollout. See Appendix G.

Figure 2 demonstrates the empirical effect of the experiments. Note that the learning and loss curves looked notably different across ten random seeds, even though the control and experimental group have the *exact same hyperparameters*. This experiment shows that IMPALA algorithmically could be susceptible to reproducibility issues across different hardware settings. While Figure 2 only shows the experimental results on one environment, the primary purpose of it is to show that this issue exists and is barely predictable. Furthermore, this type of issue can be much more subtle and difficult to diagnose at a much larger scale, so it is important that we investigate them.

## 4 TOWARDS REPRODUCIBLE DISTRIBUTED DRL

Despite these reproducibility issues, the actor-learner architecture is useful because it allows us to parallelize the computations of the actors and learners. In this work, we address the reproducibility issues mentioned above by 1) decoupling hyperparameters and hardware settings and 2) proposing a synchronization mechanism that makes distributed DRL reproducible.

### 4.1 DECOUPLING HYPERPARAMETERS AND HARDWARE SETTINGS

As mentioned in the previous section, different numbers of actor threads could make policy updates more or less frequent in the middle of a trajectory generation. This is unpredictable and need not be the case. A different number of actors also creates a different number of simulation environments and thus should be recognized as a hyperparameter setting.

To make a more clarified setting, we advocate decoupling the number of actor threads into two separate hyperparameters: 1) the number of environments, and 2) the number of CPUs. In this case, we can use a different number of CPUs to simulate a given number of environments. This decoupled interface is readily provided by EnvPool (Weng et al., 2022), which we use in our proposed architecture.

Table 1: The Synchronous and Cleanba's architecture. Under the Synchronous architecture, the actor and learner's computations are sequential and *not* parallelizable – the learner always learns from the rollout data of the latest policy $\pi_i \xrightarrow[\mathcal{D}_{\pi_i}]{} \pi_{i+1}$ (e.g., $\boxed{\pi_2 \xrightarrow[\mathcal{D}_{\pi_2}]{} \pi_3}$). Under Cleanba's architecture, we can parallelize the actor and learner's computation at the cost of introducing stale data – starting from iteration 3 the learner always learns from the rollout data obtained from the second latest policy $\pi_i \xrightarrow[\mathcal{D}_{\pi_{i-1}}]{} \pi_{i+1}$ (e.g., $\boxed{\pi_2 \xrightarrow[\mathcal{D}_{\pi_1}]{} \pi_3}$)

| Iteration | 1 | 2 | 3 |
|---|---|---|---|
| Synchronous Arch. | $\pi_1 \to \mathcal{D}_{\pi_1}$ $\pi_1 \xrightarrow[\mathcal{D}_{\pi_1}]{} \pi_2$ | $\pi_2 \to \mathcal{D}_{\pi_2}$ $\pi_2 \xrightarrow[\mathcal{D}_{\pi_2}]{} \pi_3$ | $\pi_3 \to \mathcal{D}_{\pi_3}$ $\pi_3 \xrightarrow[\mathcal{D}_{\pi_3}]{} \pi_4$ |
| Cleanba's Arch., Actor | $\pi_1 \to \mathcal{D}_{\pi_1}$ | $\pi_1 \to \mathcal{D}_{\pi_1}$ | $\pi_2 \to \mathcal{D}_{\pi_2}$ |
| Cleanba's Arch., Learner | | $\pi_1 \xrightarrow[\mathcal{D}_{\pi_1}]{} \pi_2$ | $\pi_2 \xrightarrow[\mathcal{D}_{\pi_1}]{} \pi_3$ |

## 4.2 DETERMINISTIC ROLLOUT DATA COMPOSITION

To address the non-determinism in rollout data composition, we propose our *Cleanba's architecture*, which retains the benefit of parallelizing actor-learner computations but can produce deterministic rollout data composition. At its core, Cleanba's architecture is a simple mechanism for synchronizing the actor and learner, ensuring the learner performs gradient updates with rollout data of **second latest policy**.

Let us use the notation $\pi_i \to \mathcal{D}_{\pi_i}$ to denote that policy of version $i$ is used to obtain rollout data $\mathcal{D}_{\pi_i}$; $\pi_i \xrightarrow[\mathcal{D}_{\pi_i}]{} \pi_{i+1}$ denotes policy of version $i$ is trained with rollout data $\mathcal{D}_{\pi_i}$ to obtain a new policy $\pi_{i+1}$. Figure 1 is the pseudocode of the architecture and Table 1 illustrates how policies get updated. Under the Synchronous Architecture, the actor and learner's computations are sequential: it first perform rollout $\boxed{\pi_1 \to \mathcal{D}_{\pi_1}}$, during which the learner stays idle. Given the rollout data, the learner then performs gradient updates $\boxed{\pi_1 \xrightarrow[\mathcal{D}_{\pi_1}]{} \pi_2}$, during which the actor stays idle. More generally, the learner always learns from the rollout data of the latest policy $\pi_i \xrightarrow[\mathcal{D}_{\pi_i}]{} \pi_{i+1}$.

To parallelize actor and learner's computation, Cleanba's architecture needs to necessarily introduce stale data like IMPALA (Espeholt et al., 2018). In the second iteration of Cleanba's architecture in Figure 1, we skip the `param_Q.get()` call, so $\boxed{\pi_1 \to \mathcal{D}_{\pi_1}}$ happens concurrently with $\boxed{\pi_1 \xrightarrow[\mathcal{D}_{\pi_1}]{} \pi_2}$. Because `Queue.get` is blocking when the queue is empty and `Queue.put` is blocking when the queue is full (we set the maximum size to be 1), we make sure the actor process does not perform more rollouts and learner process does not perform more gradient updates. Starting iteration $i > 3$, the learner then learns from the rollout data of the second latest policy $\pi_i \xrightarrow[\mathcal{D}_{\pi_{i-1}}]{} \pi_{i+1}$. As a result, Cleanba's architecture can parallelize the actor and learner's computation at the cost of stale data.

Cleanba's architecture above has several benefits. First, it is easy to reason and reproduce. As highlighted in Table 1, we can ascertain the specific policy used for collecting the rollout data, so if we had delayed learner updates like in Section 3 for iteration $i$, iteration $i + 1$ would not start until the previous iteration is finished, therefore circumventing IMPALA's reproducibility issue. This knowledge about which policy generates the rollout data enhances the transparency and reproducibility of distributed RL and can help us scale up while maintaining good reproducibility principles. Second, Cleanba's architecture is easy to debug for throughput. For diagnosing throughput, we can evaluate the time taken for `rollout_Q.get()` and `param_Q.get()`. If, on average, `rollout_Q.get()` consumes less time than `param_Q.get()`, it becomes evident that learning is the bottleneck, and vice versa.

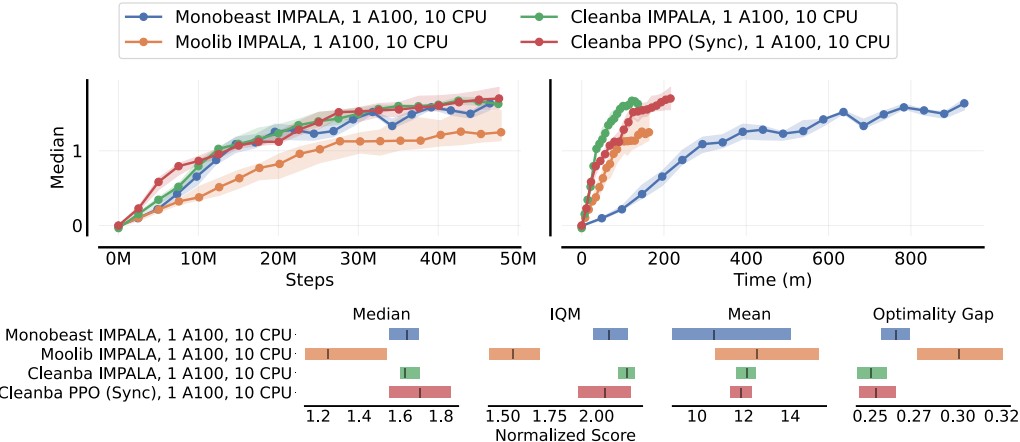

Figure 3: Base experiments. Top figure: the median human-normalized scores of Cleanba variants compared with `moolib` and `monobeast`. Bottom figure: the aggregate human normalized score metrics with 95% stratified bootstrap CIs. Higher is better for Median, IQM, and Mean; lower is better for Optimality Gap.

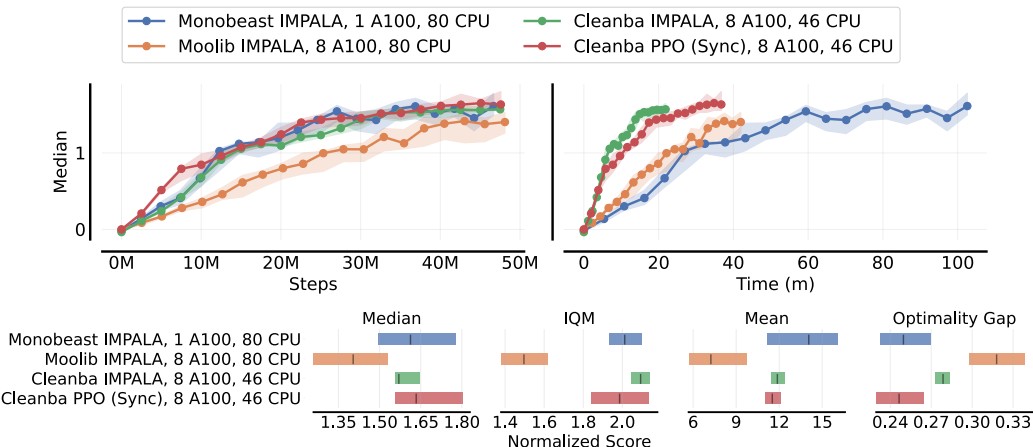

Figure 4: Workstation experiments. Top figure: the median human-normalized scores of Cleanba variants compared with `moolib`. Bottom figure: the aggregate human normalized score metrics with 95% stratified bootstrap CIs.

Based on Cleanba's architecture, this work introduces Cleanba as a reproducible distributed DRL platform. Cleanba is inspired by CleanRL (Huang et al., 2022b) and DeepMind's Sebulba Podracer architecture (Hessel et al., 2021). Its implementation uses JAX (Bradbury et al., 2018) and EnvPool (Weng et al., 2022), both of which are designed to be efficient. To improve the learner's throughput, we allow the use of multiple learner devices via `pmap`. To improve the system's scalability, we enable running multiple processes on a single node or multiple nodes via `jax.distributed`.

## 5 EXPERIMENTS

We perform experiments on Atari games (Bellemare et al., 2013). All experiments used $84 \times 84$ images with greyscale, an action repeat of 4, 4 stacked frames, and a maximum of 108,000 frames per episode. We followed the recommended Atari evaluation protocol by Machado et al. (2018), which used sticky action with a probability of 25%, no loss of life signal, and the full action space.

To make a more direct and fair comparison, we used the same AWS p4d.24xlarge instances[3] and the same Atari environment simulation setups via EnvPool and compared only the following codebase settings:

1. **Monobeast IMPALA**: the reference IMPALA implementations in `monobeast`[4];
2. **Moolib IMPALA**: the reference IMPALA implementations in `Moolib`;
3. **CleanRL PPO (Sync)**: the reference PPO implementations in CleanRL(Huang et al., 2022b);
4. **Cleanba PPO** and **Cleanba IMPALA**: our PPO and IMPALA implementation under the Cleanba Architecture;
5. **Cleanba PPO (Sync)** and **Cleanba IMPALA (Sync)** our PPO and IMPALA implementation under the Synchronous Architecture (Table 1), which can be configured by commenting out line 7 of the Cleanba's architecture in Figure 1.

Within the p4d.24xlarge instance, we also compared two hardware settings:

1. **Base experiments** uses 10 CPU and 1 A100 setting as a base comparison;
2. **Workstation experiments** uses 46 CPU and 8 A100s for Cleanba experiments, 80 CPU and 8 A100s for `moolib` experiments[5], and 80 CPU and 1 A100 for `monobeast` experiments.

Throughout all experiments, the agents used IMPALA's Resnet architecture (Espeholt et al., 2018), ran for 200M frames with three random seeds. The hyperparameters and the learning curves can be found in Appendix B. We evaluate the experiment results based on median HNS learning curves, interquartile mean (IQM) learning curves, and 95% stratified bootstrap confidence intervals for the mean, median, IQM, and optimality gap (the amount by which the algorithm fails to meet a minimum normalized score of 1) (Agarwal et al., 2021). To examine scalability in multi-node settings, we conduct experiments examining scalability on 16, 32, 64, and 128 A100s (Appendix F).

## 5.1 COMPARISON WITH MOOLIB AND MONOBEAST'S IMPALA

Under the base experiments (Figure 3), Cleanba's IMPALA obtains a similar level of median HNS as `monobeast`'s IMPALA and a higher level of median HNS as `moolib`'s IMPALA. However, Cleanba's IMPALA is **6.8x faster** than `monobeast`'s IMPALA, mostly because Cleanba actors run on GPUs, whereas `monobeast`'s actors run on CPUs. Also, Cleanba's IMPALA is **1.2x faster** than `moolib`'s IMPALA, but the speedup difference is challenging to explain due to multiple confounding factors – Cleanba's variants benefit from JAX's just-in-time compilation, whereas `moolib` benefits from asynchronous operations (e.g., on gradient computation and environment steps). Cleanba's PPO (Sync) also obtains a high median HNS but takes longer training time, likely due to the longer training step time spent on reusing rollout data 4 times.

Under the workstation experiments (Figure 4), Cleanba's PPO (Sync) and IMPALA obtain a similar level of median HNS as `monobeast`'s IMPALA and a higher level of median HNS as `moolib`'s IMPALA. However, Cleanba's PPO (Sync) and IMPALA are both faster than `monobeast`'s and `moolib` IMPALA. Most prominently, Cleanba's IMPALA is **5x faster** than `monobeast`'s IMPALA and **2x faster** than `moolib`'s IMPALA.

Additionally, we examine the individual learning curves in Figure 5 and found that Cleanba's variants also produce more consistent learning curves. In comparison, in two hardware settings, `moolib`'s learning curves can be much more unpredictable.

---

[3]For some experiments, we used p4de.24xlarge instances but only GPU memory is different, which does not affect training speed.

[4]We wanted to test out IMPALA's official source code released in `deepmind/scalable_agent`, but it was built with `tensorflow 1.x` which does not support the A100 GPU tested in this paper.

[5]We used more CPUs for `moolib` experiments because 10 CPU per GPU seems to be the default scaling parameter for `moolib`. Also, for the `moolib` experiment, we conducted two sets of 3 random seeds. We reported the results with higher IQM and lower median. See Appendix C.

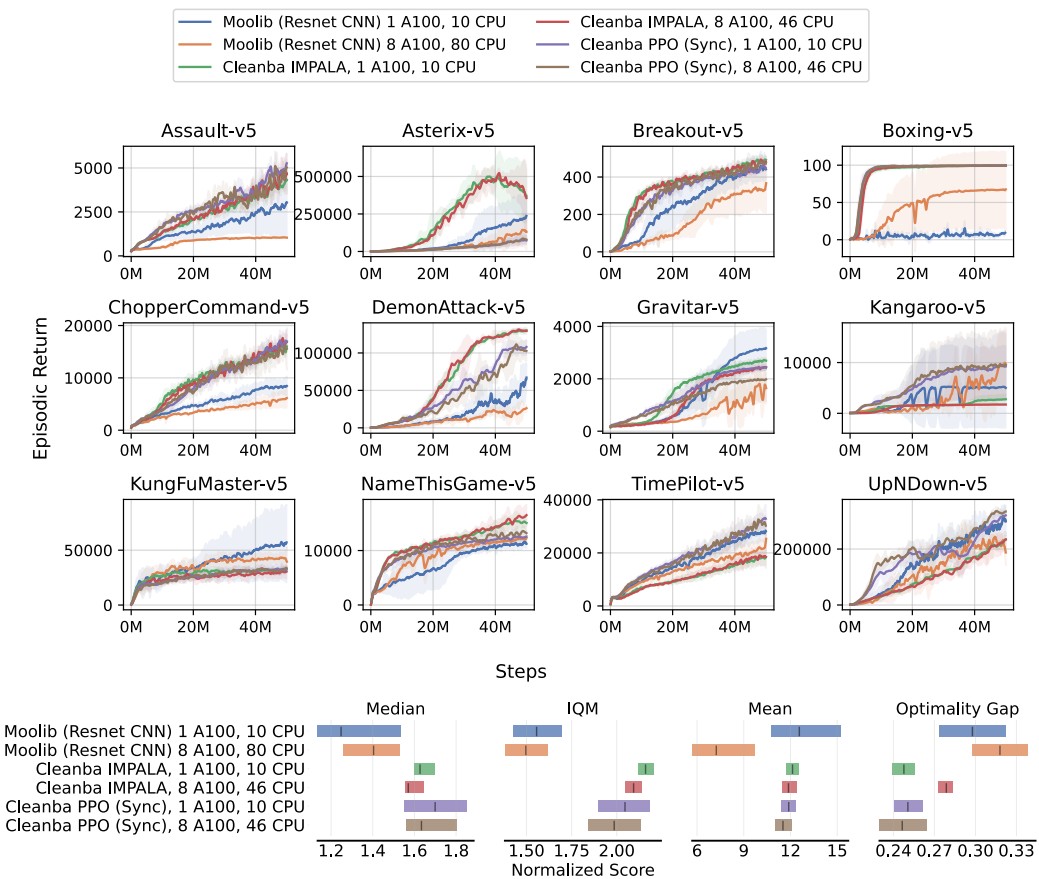

Figure 5: Reproducible learning curves – the Cleanba variants show more predictable learning curves in different hardware settings. In comparison, `moolib`'s IMPALA's learning curves under the 1 A100, 10 CPU setting (blue curve) and 8 A100, 80 CPU setting (orange curve) are meaningfully different, even if they use the same hyperparameters.

## 5.2 DISCUSSION ABOUT MONOBEAST'S IMPALA

Note that the `monobeast` experiments are interesting in several ways. First, it produces a higher median HNS than `moolib`'s IMPALA, which is the opposite of what was shown in Mella et al. (2022). This is probably because Mella et al. (2022) used "comparable environment settings" instead of the same environment settings used in our experiments. Interestingly, we found different Atari wrapper implementations can have a non-trivial impact on the agent's performance (Appendix D); for this reason, we use the same Atari wrapper implementation in the experiments presented in this section. Second, the `monobeast` experiments appear robust in two different hardware settings in practice, despite the reproducibility issues we showed in Section 3. While `monobeast` obtained high scores, it is significantly slower in the 1 A100 and 10 CPU settings due to poor GPU utilization. Its codebase also does not support multi-GPU settings and should scale less efficiently with larger networks because actor threads only run on CPUs when compared to `moolib` and Cleanba's variants.

## 5.3 SYNCHRONOUS ARCHITECTURE VS CLEANBA ARCHITECTURE

Figure 6 compares the PPO and IMPALA variants between Synchronous and Cleanba architecture and CleanRL's PPO, which uses the Synchronous architecture by design. We found using Cleanba architecture actually hurts Cleanba PPO's data efficiency. This is an interesting trade-off because the speed benefit of parallelizing actor and learner processes in Cleanba PPO is offset by the lower

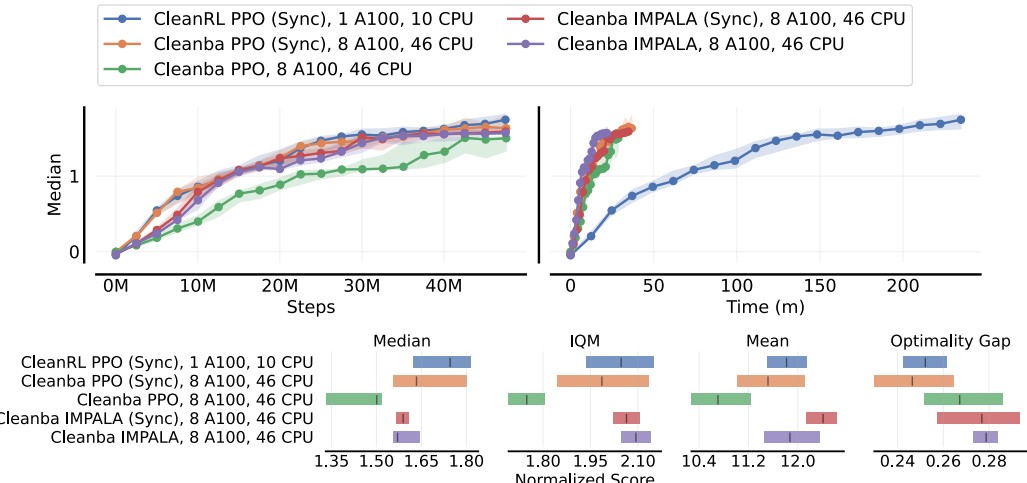

Figure 6: Comparing Cleanba's variants using Cleanba and Synchronous architecture. For PPO, Cleanba's Architecture (orange curve) runs faster but has lower data efficiency than Synchronous architecture (blue curve). For IMPALA, there is no discernible difference between Synchronous Architecture (red curve) and Cleanba's Architecture (brown curve). This means Cleanba's IMPALA can benefit from the speed-up of parallelizing actor-learner computation without paying a price for data efficiency under our hyperparameter settings, unlike Cleanba's PPO.

data efficiency. Among many possible causes, the main factor might be that PPO does 16 gradient updates (4 mini-batches and 4 update epochs) per rollout, whereas IMPALA in our setting only does 4 gradient updates. In comparison, we noticed Cleanba's IMPALA did not suffer from lower data efficiency compared to Cleanba IMPALA (Sync) architecture, meaning IMPALA can actually benefit from parallelizing actor and learner computations.

## 6  LIMITATION

There are several limitations to this work. First, our experiments could not completely control various other confounding settings in the reference codebase, such as optimizer settings and machine learning framework (e.g., PyTorch, JAX). For example, Cleanba's PPO and IMPALA use different learning rates indicated in their respective literature, making it difficult to compare PPO and IMPALA directly. We attempted to make a direct comparison by running Cleanba PPO with Cleanba IMPALA's setting and found it made PPO's data efficiency significantly worse – this could suggest the IMPALA's setting is well-tuned for IMPALA but brittle to PPO (Appendix E). Second, our finding that parallelizing actor and learner computation hurts PPO's data efficiency is specific to the PPO's default Atari hyperparameter setting, and it could perhaps be tuned in ways in which opposite findings can be drawn. That said, the main purpose of this work is not hyperparameter tuning. Rather, it is creating a codebase that replicates prior results and makes training reproducible, efficient, and scalable across more powerful hardware.

## 7  CONCLUSION

This paper presents Cleanba, a new distributed deep reinforcement learning platform. Our analysis shows that Cleanba's more principled architecture can circumvent reproducibility issues in IMPALA's architecture. Our Atari experiments demonstrate that Cleanba's PPO and IMPALA accurately replicate prior work but have faster training time and are highly reproducible across different hardware settings. We believe that Cleanba will be a valuable platform for the research community to conduct future distributed RL research.

## ACKNOWLEDGMENTS

We thank the following entities for their support.

1. Stability AI's HPC for generously providing much GPU computational resources to this project.
2. Hugging Face's cluster for providing much GPU computational resources to this project.
3. Google's TPU Research Cloud for providing the TPU computational resources.

## REPRODUCIBILITY STATEMENT

Ensuring Cleanba's results are reproducible is a central theme in our paper. To this end, we have taken several measures to improve reproducibility:

1. **Open-source repository**: we made source code available at `https://github.com/vwxyzjn/cleanba`. The dependencies of the experiments are pinned, and our repository contains detailed instructions on replicating all Cleanba experiments presented in this paper.
2. **Reproducible architecture**: as demonstrated in Section 4, Cleanba introduces a more principled approach to understanding distributed DRL and gives clear expectations on where the rollout data comes from, making it easier to reason about the reproducibility of distributed DRL.
3. **Experiments on different hardware**: as demonstrated in Section 5, we also conducted experiments showing Cleanba's PPO and IMPALA variants can obtain near-identical data efficiency on different hardware, further demonstrating that this work is highly reproducible.

In sum, we have tried to make our work as transparent and reproducible as possible. By leveraging the source code, details provided in the main paper, and appendix, researchers should be well-equipped to reproduce or extend upon our findings.

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
