---

**Algorithm 1** Proximal Policy Optimization

---

1: **Initialize** environment $E$ containing `local_num_envs` parallel sub-environments
2: **Initialize** policy parameters $\theta_\pi$, value parameters $\theta_v$, optimizer $O$
3: **Initialize** observation $s_{next}$, done flag $d_{next}$
4: **for** $i = 0,1,2,..., I$ **do**
5:     **Set** $\mathcal{D} = (s, a, \log \pi(a|s), r, d, v)$ as tuple of 2D arrays
6:     **for** $t = 0,1,2,...,$ `num_steps` **do**                          ▷ Rollout Phase
7:         **Cache** $o_t = s_{next}$ and $d_t = d_{next}$
8:         **Get** $a_t \sim \pi(\cdot|s_t; \theta_\pi)$ and $v_t = v(s_t; \theta_v)$
9:         **Step** simulator: $s_{next}, r_t, d_{next} = E.step(a_t)$
10:        **Store** $s_t, d_t, v_t, a_t, \log \pi(a_t|s_t; \theta_\pi), r_t$ in $\mathcal{D}$
11:    **Estimate** next value $v_{next} = v(s_{next})$                    ▷ Learning Phase
12:    **Compute** advantage $\hat{A}_\pi^{adv}$ and return $R$ using $\mathcal{D}$ and $v_{next}$
13:    **Prepare** the batch $\mathcal{B} = \mathcal{D}, \hat{A}_\pi^{adv}, R$ and flatten $\mathcal{B}$
14:    **for** $epoch = 0,1,2,...,$ `update_epochs` **do**
15:        **for** mini-batch $\mathcal{M}$ of size $m$ in $\mathcal{B}$ **do**
16:            **Normalize** advantage $\mathcal{M}.A_\pi^{adv}$
17:            **Compute** policy loss $L^\pi$, value loss $L^V$, and entropy loss $L^S$ using $\mathcal{M}$
18:            **Back-propagate** joint loss $L = -L^\pi + c_1 L^V - c_2 L^S$
19:            **Clip** maximum gradient norm of $\theta_\pi$ and $\theta_v$ to 0.5
20:            **Step** optimizer $O$ w.r.t. $\theta_\pi$ and $\theta_v$

---

## A    PRELIMINARIES

Let us consider the RL problem in a *Markov Decision Process (MDP)* (Puterman, 2014), where $\mathcal{S}$ is the state space and $\mathcal{A}$ is the action space. The agent performs some actions to the environment, and the environment transitions to another state according to its *dynamics* $P(s' \mid s,a) : \mathcal{S} \times \mathcal{A} \times \mathcal{S} \rightarrow [0, 1]$. The environment also provides a scalar reward according to the reward function $R : \mathcal{S} \times \mathcal{A} \rightarrow \mathbb{R}$, and the agent attempts to maximize the expected discounted return following a policy $\pi$:

$$J(\pi) = \mathbb{E}_\tau \left[ G(\tau) \right] \tag{1}$$

where $\tau$ is the trajectory $(s_0, a_0, r_0, \ldots, s_{T-1}, a_{T-1}, r_{T-1}, s_T)$
and $s_0 \sim \rho_0, s_t \sim P(\cdot|s_{t-1}, a_{t-1}), a_t \sim \pi_\theta(\cdot|s_t), r_t = r(s_t, a_t)$

PPO (Schulman et al., 2017) is a popular algorithm that proposes a clipped policy gradient objective to help avoid unstable updates (Schulman et al., 2017; 2015):

$$J^{CLIP}(\pi_\theta) = \mathbb{E}_\tau \left[ \sum_{t=0}^{T-1} \min \left( r_t(\theta) \hat{A}_\pi^{adv}(s_t, a_t), \text{clip}\left( r_t(\theta), 1 - \epsilon, 1 + \epsilon \right) \hat{A}_\pi^{adv}(s_t, a_t) \right) \right] \tag{2}$$

where $\pi_{\theta_{old}}$ is the policy parameter before the update, $r_t(\theta) = \frac{\pi_\theta(a_t|s_t)}{\pi_{\theta_{old}}(a_t|s_t)}$, $\hat{A}_\pi^{adv}$ is an advantage estimator called Generalized Advantage Estimator (Schulman et al., 2016), and $\epsilon$ is PPO's clipped coefficient. During the optimization phase, the agent also learns the value function and maximizes the policy's entropy, therefore optimizing the following joint objective:

$$J^{JOINT}(\theta) = J^{CLIP}(\pi_\theta) - c_1 J^{VF}(\theta) + c_2 S[\pi_\theta], \tag{3}$$

where $c_1, c_2$ are coefficients, $S$ is an entropy bonus, and $J^{VF}$ is the squared error loss for the value function associated with $\pi_\theta$. Algorithm 1 shows the pseudocode of PPO that more accurately reflects how PPO is implemented in the original codebase[6]. For more detail on PPO's implementation, see (Huang et al., 2022a). Given this pseudocode, the following list unifies the nomenclature/terminology of PPO's key hyperparameters.

- `world_size` is the number of instances of training processes; typically this is 1 (e.g., you have a single GPU).

---

[6] https://github.com/openai/baselines

- `local_num_envs` is the number of parallel environments PPO interacts within an instance of the training process (see line 1). `num_envs = world_size * local_num_envs` is the total number of environments across all training instances.

- `num_steps` is the number of steps in which the agent samples a batch of `local_num_envs` actions and receives a batch of `local_num_envs` next observations, rewards, and done flags from the simulator (see line 6), where the done flags signal if the episodes are terminated or truncated. `num_steps` has many names, such as the "sampling horizon" (Stooke & Abbeel, 2018) and "unroll length" (Freeman et al., 2021).

- `local_batch_size` is the batch size calculated as `local_num_envs * num_steps` within an instance of the training process (`local_batch_size` is the size of the $\mathcal{B}$ in line 13).

- `batch_size = world_size * local_batch_size` is the aggregated batch size across all training instances.

- `update_epochs` is the number of update epochs that the agent goes through the training data in $\mathcal{B}$ (see line 14).

- `num_minibatches` is the number of mini-batches that PPO splits $\mathcal{B}$ into (see line 15).

- `local_minibatch_size` is $m = $ `local_batch_size / num_minibatches`, the size of each mini-batch $\mathcal{M}$ (see line 15). `minibatch_size = world_size * local_minibatch_size` is the aggregated batch size across all training instances.

To make understanding more concrete, let us consider an example of Atari training. Typically, PPO uses a single training instance (i.e., `world_size` = 1), `local_num_envs` = `num_envs` = 8, and `num_steps` = 128. In the rollout phase (line 6-10), the agent collects a batch of $8 * 128 = 1024$ data points in $\mathcal{D}$. Then, suppose `num_minibatches` = 4, $\mathcal{D}$ is evenly split to 4 mini-batches of size $m = 1024/4 = 256$. Next, if $K = 4$, the agent would perform $K *$ `num_minibatches` = 16 gradient updates in the learning phase (line 11-20).

We consider two options to scale to larger training data. **Option 1** is to increment `local_num_envs` – the agent interacts with more environments, and as a result, the training data is larger. The second option is to increment `world_size` – have two or more copies of Algorithm 1 running in parallel and average the gradient of the copies in line 20. **Option 2** is especially desirable when the users want to leverage more computational resources, such as GPUs.

Note that both options can be equivalent *in terms of hyperparameters*. For example, when setting `world_size` = 2, the agent effectively interacts with two distinct sets of `local_num_envs` environments, making its `num_envs` doubled. To make option 1 achieve the same hyperparameters, we just need to double its `local_num_envs`. Below is a table summarizing the resulting hyperparameters of both options.

| Hyperparameter | Option 1: Increment `local_num_envs` | Option 2: Increment `world_size` |
|---|---|---|
| `world_size` | 1 | 2 |
| `local_num_envs` | 120 | 60 |
| `num_envs` | **120** | **120** |
| `num_steps` | 128 | 128 |
| `local_batch_size` | 15360 | 7680 |
| `batch_size` | **15360** | **15360** |
| `num_minibatches` | 4 | 4 |
| `local_minibatch_size` | 3840 | 1920 |
| `minibatch_size` | **3840** | **3840** |

Importantly, we can get the same hyperparameter configuration for PPO by adjusting `local_num_envs` and `world_size` accordingly. That is, we can obtain the same `num_envs`, `batch_size`, and `minibatch_size` core hyperparameters.

# B    DETAILED EXPERIMENT SETTINGS

For the experiments, the PPO and IMPALA's hyperparameters can be found in Table 2. The Vtrace implementation can be found in `rlax`[7].

Table 2: PPO hyperparameters.

| Parameter Names | Parameter Values |
| --- | --- |
| $N_{\text{total}}$ Total Time Steps | 50,000,000 |
| $\alpha$ Learning Rate | 0.00025 Linearly Decreased to 0 |
| $N_{\text{envs}}$ Number of Environments | 128 |
| $N_{\text{steps}}$ Number of Steps per Environment | 128 |
| $\gamma$ (Discount Factor) | 0.99 |
| $\lambda$ (for GAE) | 0.95 |
| $N_{\text{mb}}$ Number of Mini-batches | 4 |
| $K$ (Number of PPO Update Iteration Per Epoch) | 4 |
| $\varepsilon$ (PPO's Clipping Coefficient) | 0.1 |
| $c_1$ (Value Function Coefficient) | 0.5 |
| $c_2$ (Entropy Coefficient) | 0.01 |
| $\omega$ (Gradient Norm Threshold) | 0.5 |
| Value Function Loss Clipping | False |
| Optimizer Setting | Adam optimizer with $\epsilon = 0.00001$ |

Table 3: IMPALA hyperparameters.

| Parameter Names | Parameter Values |
| --- | --- |
| $N_{\text{total}}$ Total Time Steps | 50,000,000 |
| $\alpha$ Learning Rate | 0.0006 Linearly Decreased to 0 |
| $N_{\text{envs}}$ Number of Environments | 128 |
| $N_{\text{steps}}$ Number of Steps per Environment | 20 |
| $\gamma$ (Discount Factor) | 0.99 |
| $\lambda$ (mixing parameter) | 1.0 |
| $N_{\text{mb}}$ Number of Mini-batches | 4 |
| $\rho$ (Clip Threshold for Importance Ratios) | 1.0 |
| $\rho_{pg}$ (Clip Threshold for Policy Gradient Importance Ratios) | 1.0 |
| $c_1$ (Value Function Coefficient) | 0.5 |
| $c_2$ (Entropy Coefficient) | 0.01 |
| $\omega$ (Gradient Norm Threshold) | 40.0 |
| Optimizer Setting | RMSprop optimizer with $\epsilon = 0.01$, decay $= 0.99$ |

# C    MOOLIB EXPERIMENTS

By default, `moolib` uses 256 environments, 10 actor CPUs, and a single GPU. We followed the recommended scaling instructions to add 8 training GPU-powered peers, which in total used 2048 environments, 80 actor CPUs, and 8 GPUs. While the training time was reduced to about 27 minutes, sample efficiency dropped, and it obtained a catastrophic 28.51% median HNS after 200M frames. We suspected the drop was due to the 2048 environments used, so we set the total number of environments back to 256. Furthermore, we did not restrict `moolib` to use 50 CPUs because we worried it might change the learning behaviors due to the issues mentioned in Section 3, so we kept the default scaling to 80 CPUs. For comparison with `moolib`, `monobeast` experiments also use 80 CPUs.

---

[7]https://github.com/deepmind/rlax/blob/b53c6510c8b2cad6b106b6166e22aba61a77ee2f/rlax/_src/vtrace.py#L162-L193

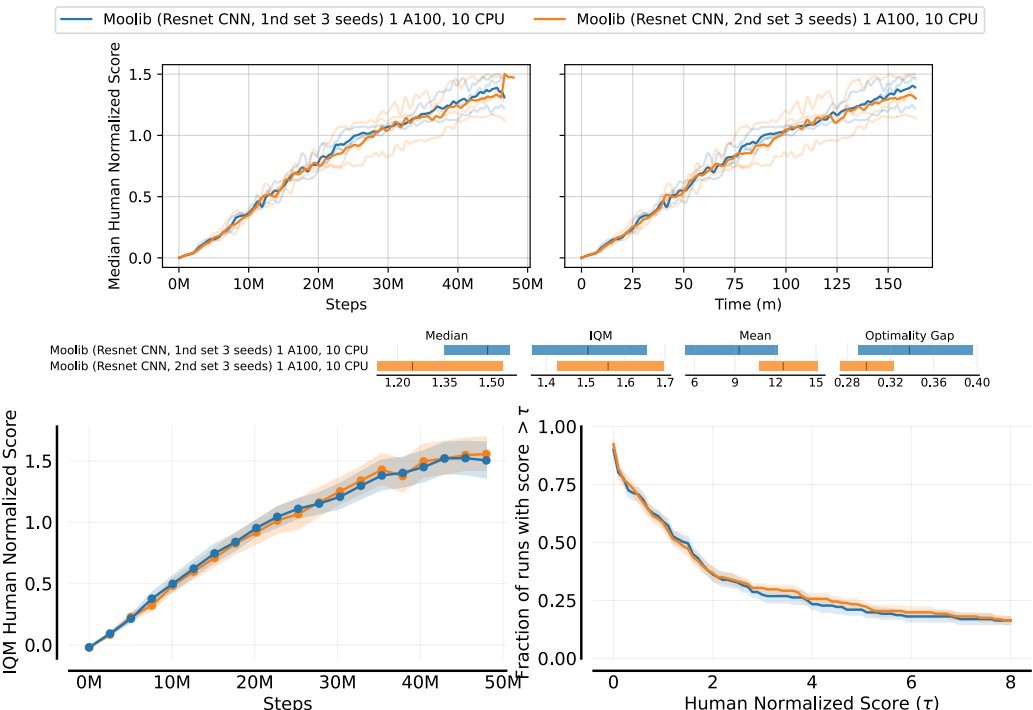

Figure 7: Top figure: the median human-normalized scores of the two sets of `moolib` experiments. Middle figure: the IQM human-normalized scores and performance profile (Agarwal et al., 2021). Bottom figure: the average runtime in minutes and aggregate human normalized score metrics with 95% stratified bootstrap CIs.

We conducted two sets of `moolib` experiments and reported the set with a lower median and higher IQM, as shown in Figure 7 for legacy reasons. During our debugging, we found the Asteroids experiments in the first set of `moolib` experiments to obtain high scores, but we ran Asteroids specifically for ten random seeds and found lower scores; this suggests the Asteroids experiments in the first set were likely due to lucky random seeds, so we re-run the `moolib` experiments.

## D   THE EFFECT OF DIFFERENT WRAPPERS ON MOOLIB'S PERFORMANCE

The Atari wrappers can be important to the agent's performance. As a preliminary study, we used `moolib`'s default Atari wrappers[8] implemented with `gym.AtariPreprocessing` to run experiments and compare the results with the ones presented in the main text of the paper. As shown in Figure 8, Atari wrappers matter – `moolib`'s default `AtariPreprocessing` wrappers result in lower median and mean HNS, although IQM is roughly the same. To make a fair comparison, the experiments presented in the main text all use the same EnvPool Atari wrappers.

## E   DIRECT PPO AND IMPALA COMPARISON

To make a direct (but not fair) comparison between PPO and IMPALA, we ran Cleanba PPO using IMPALA's settings and the results can be found at Figure 9.

---

[8]https://github.com/facebookresearch/moolib/blob/main/examples/atari/atari_preprocessing.py

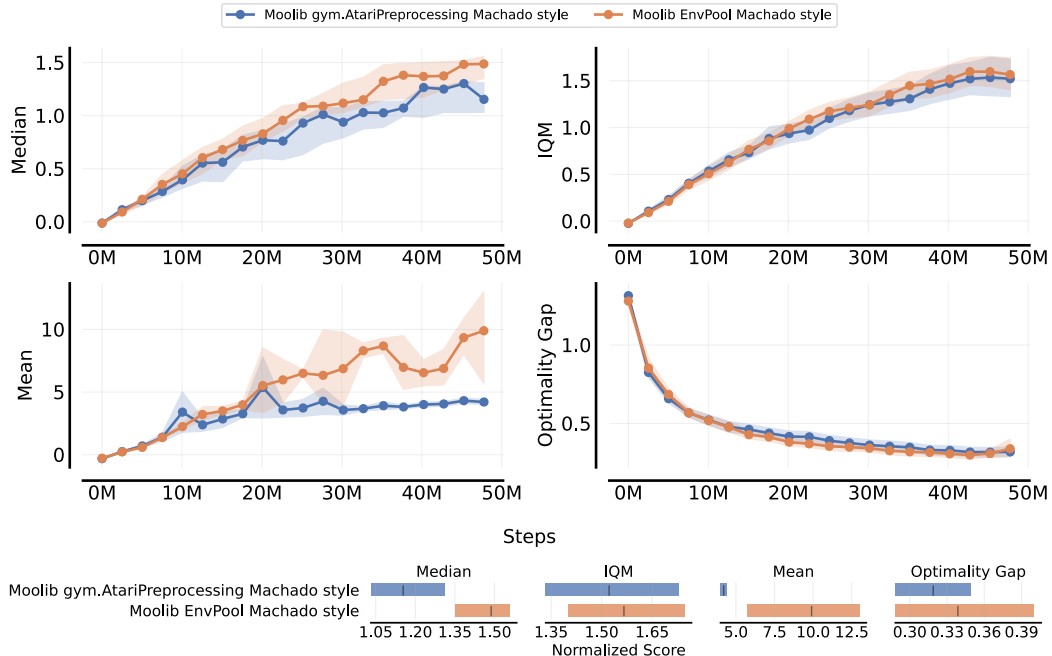

Figure 8: Atari wrappers matter. When using `gym.AtariPreprocessing` wrappers with a comparable setting to our EnvPool setup, we found `moolib`to have lower median and mean HNS, although IQM is roughly the same.

## F  MULTI-NODE TRAINING

Cleanba can also scale to the hundreds of GPUs in multi-host and multi-process environments by leveraging the `jax.distributed` package, allowing us to explore training with even larger batch sizes. Here we use an earlier version of the codebase to conduct experiments with 16, 32, 64, and 128 A100 GPUs. We keep doubling the `num_envs`, `batch_size`, and `minibatch_size` with a larger number of GPUs (except for the 128 A100 experiment which uses the same `num_envs` as the 64 A100 experiment accidentally).

Due to hardware scheduling constraints, we only ran the experiments for 1 random seed. The results are shown in Figure 10. We make the following observations:

- **Linear scaling w/ 97% of ideal scaling efficiency.** As we increased the number of GPUs to 16, 32, 64, we observed a linear scaling in steps per second (SPS) in Cleanba achieving $1013163/(258754 * 4) = 97\%$ of the ideal scaling efficiency. This is likely empowered by the fast connectivity offered by NVIDIA GPUDirect RDMA (remote direct memory access) in our cluster. Note that the 128 A100 experiments did not double the batch size so its FPS is lower.

- **Small batch sizes train more efficiently.** As we increase batch sizes, particularly in the first 40M steps, the sample efficiency tends to decline. This outcome is unsurprising, given that the initial policy is random and Breakout initially has limited explorable game states. In this case, the data in the batch is going to have less diverse data, which makes the large batch size less valuable.

- **Large batch sizes train more quickly.** Like (McCandlish et al., 2018), we find increasing the batch size does make the agent reach some given scores faster. This suggests that we could always increase the batch size to obtain shorter training times if sample efficiency is not a concern.

While we observed limited benefits of scaling Cleanba to use 128 GPUs, the objective of the scaling experiments is to show we can scale to large batch sizes. Given a more challenging task, the training

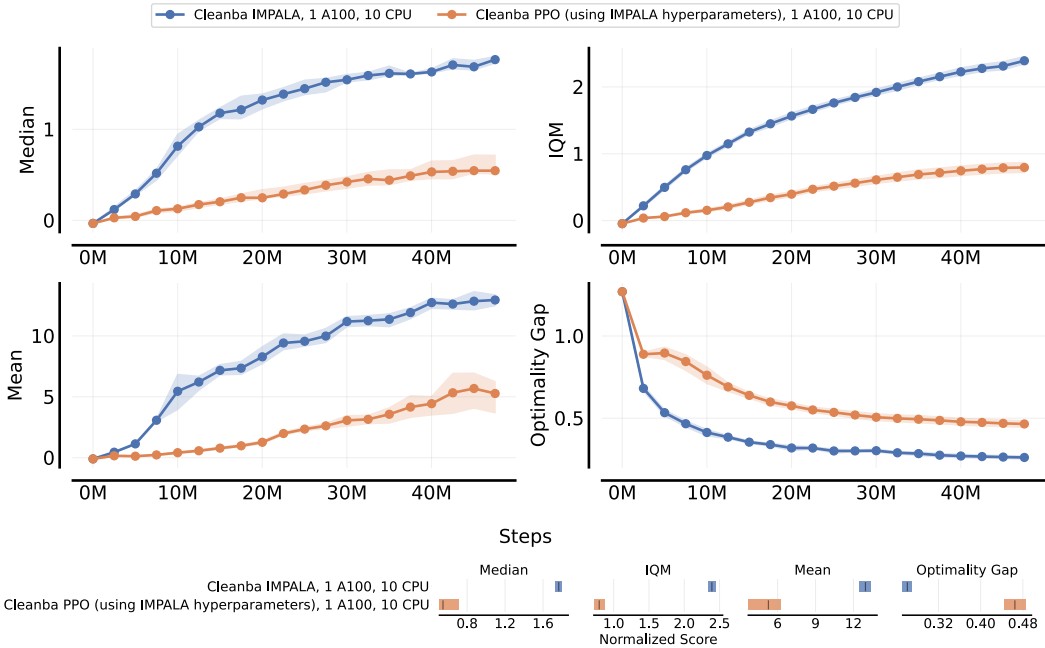

Figure 9: A direct PPO and IMPALA comparison. Running Cleanba PPO using Cleanba IMPALA's setting. Note that this is not a fair comparison because Cleanba IMPALA's setting is likely well-tuned IMPALA setting but not well-tuned PPO setting

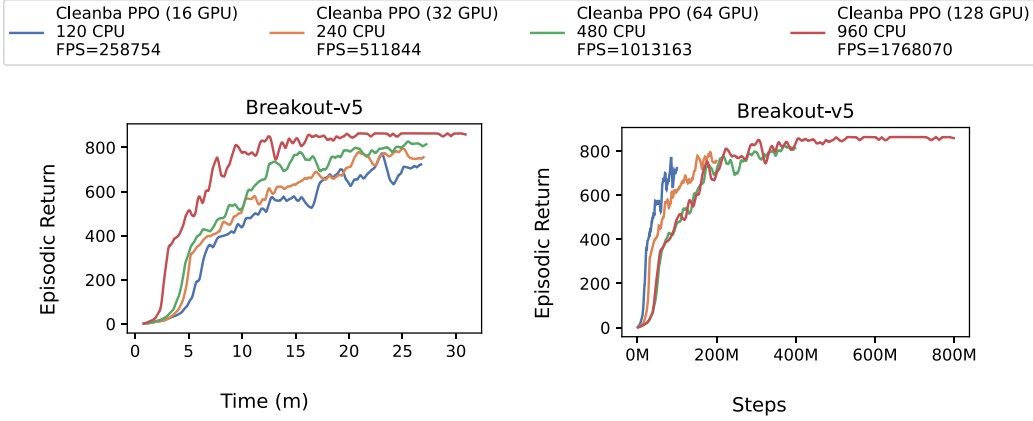

Figure 10: Cleanba's results from large batch size training. Going from 16 GPUs to 128 GPUs.

data is likely going to be more diverse and have a higher *gradient noise scale* (McCandlish et al., 2018), which would help the agent utilize large batch sizes more efficiently, resulting in a reduced decline in sample efficiency.

## G TORCHBEAST LOGS

```
$ python -m torchbeast.monobeast_study \
--num_actors 80 \
--total_steps 10000000 \
--learning_rate 0.0006 \
```

```
--epsilon 0.01 \
--entropy_cost 0.01 \
--batch_size 8 \
--unroll_length 240 \
--num_threads 1 \
--env Pong-v5
actor_index 32 initial policy_version 8 policy_version after rollout 20
actor_index 13 initial policy_version 8 policy_version after rollout 20
actor_index 57 initial policy_version 8 policy_version after rollout 20
actor_index 12 initial policy_version 8 policy_version after rollout 21
actor_index 51 initial policy_version 8 policy_version after rollout 21
actor_index 2 initial policy_version 8 policy_version after rollout 21
actor_index 56 initial policy_version 8 policy_version after rollout 21
actor_index 38 initial policy_version 9 policy_version after rollout 21
actor_index 37 initial policy_version 9 policy_version after rollout 22
actor_index 59 initial policy_version 9 policy_version after rollout 22
actor_index 9 initial policy_version 9 policy_version after rollout 22
actor_index 69 initial policy_version 9 policy_version after rollout 22
actor_index 35 initial policy_version 9 policy_version after rollout 22
actor_index 66 initial policy_version 9 policy_version after rollout 22
actor_index 10 initial policy_version 9 policy_version after rollout 22
actor_index 55 initial policy_version 10 policy_version after rollout 22
actor_index 53 initial policy_version 10 policy_version after rollout 22
actor_index 46 initial policy_version 10 policy_version after rollout 22
actor_index 54 initial policy_version 10 policy_version after rollout 23
actor_index 50 initial policy_version 10 policy_version after rollout 23
actor_index 8 initial policy_version 10 policy_version after rollout 23
actor_index 64 initial policy_version 10 policy_version after rollout 23
actor_index 77 initial policy_version 10 policy_version after rollout 23
actor_index 3 initial policy_version 11 policy_version after rollout 23
actor_index 7 initial policy_version 11 policy_version after rollout 23
actor_index 28 initial policy_version 11 policy_version after rollout 23
actor_index 49 initial policy_version 11 policy_version after rollout 23
actor_index 16 initial policy_version 11 policy_version after rollout 23
actor_index 24 initial policy_version 11 policy_version after rollout 23
actor_index 11 initial policy_version 11 policy_version after rollout 23
actor_index 14 initial policy_version 11 policy_version after rollout 23
actor_index 43 initial policy_version 13 policy_version after rollout 26
actor_index 58 initial policy_version 13 policy_version after rollout 26
actor_index 23 initial policy_version 13 policy_version after rollout 26
actor_index 29 initial policy_version 13 policy_version after rollout 26
actor_index 68 initial policy_version 13 policy_version after rollout 26
actor_index 75 initial policy_version 14 policy_version after rollout 26
actor_index 48 initial policy_version 14 policy_version after rollout 27
actor_index 67 initial policy_version 14 policy_version after rollout 27
actor_index 5 initial policy_version 14 policy_version after rollout 27
actor_index 18 initial policy_version 14 policy_version after rollout 27
actor_index 41 initial policy_version 15 policy_version after rollout 27
actor_index 78 initial policy_version 14 policy_version after rollout 27
actor_index 15 initial policy_version 15 policy_version after rollout 27
actor_index 34 initial policy_version 15 policy_version after rollout 27
actor_index 45 initial policy_version 15 policy_version after rollout 28
actor_index 22 initial policy_version 15 policy_version after rollout 28
actor_index 4 initial policy_version 16 policy_version after rollout 28
actor_index 6 initial policy_version 16 policy_version after rollout 28
actor_index 20 initial policy_version 16 policy_version after rollout 28
actor_index 39 initial policy_version 16 policy_version after rollout 28
actor_index 33 initial policy_version 16 policy_version after rollout 29
actor_index 74 initial policy_version 16 policy_version after rollout 29
actor_index 60 initial policy_version 16 policy_version after rollout 29
actor_index 42 initial policy_version 17 policy_version after rollout 29
actor_index 72 initial policy_version 17 policy_version after rollout 30
actor_index 25 initial policy_version 17 policy_version after rollout 30
actor_index 31 initial policy_version 17 policy_version after rollout 30
actor_index 19 initial policy_version 17 policy_version after rollout 30
actor_index 1 initial policy_version 18 policy_version after rollout 31
actor_index 79 initial policy_version 18 policy_version after rollout 31
```

```
actor_index 65 initial policy_version 18 policy_version after rollout 31
actor_index 73 initial policy_version 18 policy_version after rollout 31
actor_index 36 initial policy_version 18 policy_version after rollout 31
actor_index 21 initial policy_version 18 policy_version after rollout 31
actor_index 0 initial policy_version 18 policy_version after rollout 31
actor_index 30 initial policy_version 18 policy_version after rollout 31
actor_index 44 initial policy_version 18 policy_version after rollout 31
actor_index 63 initial policy_version 19 policy_version after rollout 31
actor_index 76 initial policy_version 19 policy_version after rollout 32
actor_index 47 initial policy_version 19 policy_version after rollout 32
actor_index 52 initial policy_version 19 policy_version after rollout 32
actor_index 26 initial policy_version 19 policy_version after rollout 32
actor_index 71 initial policy_version 19 policy_version after rollout 32
actor_index 70 initial policy_version 19 policy_version after rollout 32
actor_index 17 initial policy_version 20 policy_version after rollout 32
actor_index 62 initial policy_version 20 policy_version after rollout 33
actor_index 40 initial policy_version 20 policy_version after rollout 33
actor_index 27 initial policy_version 20 policy_version after rollout 33
actor_index 13 initial policy_version 20 policy_version after rollout 33
actor_index 57 initial policy_version 20 policy_version after rollout 33
actor_index 32 initial policy_version 20 policy_version after rollout 33
actor_index 51 initial policy_version 21 policy_version after rollout 33
actor_index 61 initial policy_version 20 policy_version after rollout 33
actor_index 2 initial policy_version 21 policy_version after rollout 33
actor_index 56 initial policy_version 21 policy_version after rollout 34
actor_index 12 initial policy_version 21 policy_version after rollout 34

$ python -m torchbeast.monobeast_study \
--num_actors 80 \
--total_steps 10000000 \
--learning_rate 0.0006 \
--epsilon 0.01 \
--entropy_cost 0.01 \
--batch_size 8 \
--unroll_length 240 \
--num_threads 1 \
--env Pong-v5  \
--learner_delay_seconds 1.0

actor_index 72 initial policy_version 9 policy_version after rollout 10
actor_index 22 initial policy_version 9 policy_version after rollout 10
actor_index 37 initial policy_version 9 policy_version after rollout 10
actor_index 41 initial policy_version 9 policy_version after rollout 10
actor_index 16 initial policy_version 9 policy_version after rollout 10
actor_index 61 initial policy_version 10 policy_version after rollout 11
actor_index 18 initial policy_version 10 policy_version after rollout 11
actor_index 13 initial policy_version 10 policy_version after rollout 11
actor_index 56 initial policy_version 10 policy_version after rollout 11
actor_index 28 initial policy_version 10 policy_version after rollout 11
actor_index 4 initial policy_version 10 policy_version after rollout 11
actor_index 7 initial policy_version 10 policy_version after rollout 11
actor_index 65 initial policy_version 10 policy_version after rollout 11
actor_index 12 initial policy_version 11 policy_version after rollout 12
actor_index 14 initial policy_version 11 policy_version after rollout 12
actor_index 5 initial policy_version 11 policy_version after rollout 12
actor_index 3 initial policy_version 11 policy_version after rollout 12
actor_index 35 initial policy_version 11 policy_version after rollout 12
actor_index 51 initial policy_version 11 policy_version after rollout 12
actor_index 0 initial policy_version 11 policy_version after rollout 12
actor_index 6 initial policy_version 11 policy_version after rollout 12
actor_index 60 initial policy_version 12 policy_version after rollout 13
actor_index 77 initial policy_version 12 policy_version after rollout 13
actor_index 48 initial policy_version 12 policy_version after rollout 13

$ python -m torchbeast.monobeast_study \
--num_actors 40 \
```

```
--total_steps 10000000 \
--learning_rate 0.0006 \
--epsilon 0.01 \
--entropy_cost 0.01 \
--batch_size 8 \
--unroll_length 240 \
--num_threads 1 \
--env Pong-v5

actor_index 34 initial policy_version 12 policy_version after rollout 18
actor_index 25 initial policy_version 13 policy_version after rollout 18
actor_index 4 initial policy_version 13 policy_version after rollout 18
actor_index 5 initial policy_version 13 policy_version after rollout 18
actor_index 14 initial policy_version 13 policy_version after rollout 18
actor_index 16 initial policy_version 13 policy_version after rollout 18
actor_index 12 initial policy_version 13 policy_version after rollout 18
actor_index 39 initial policy_version 13 policy_version after rollout 18
actor_index 30 initial policy_version 13 policy_version after rollout 18
actor_index 18 initial policy_version 13 policy_version after rollout 18
actor_index 13 initial policy_version 13 policy_version after rollout 18
actor_index 23 initial policy_version 13 policy_version after rollout 19
actor_index 35 initial policy_version 13 policy_version after rollout 19
actor_index 3 initial policy_version 14 policy_version after rollout 19
actor_index 17 initial policy_version 14 policy_version after rollout 19
actor_index 9 initial policy_version 14 policy_version after rollout 19
actor_index 6 initial policy_version 14 policy_version after rollout 19
```