# OpenReview forum: "Cleanba: A Reproducible and Efficient Distributed Reinforcement Learning Platform"
_ICLR.cc/2024/Conference — ICLR 2024 poster_

### Official Review · Reviewer_eKtC · 2023-10-31

**Soundness:** 3 good
**Presentation:** 3 good
**Contribution:** 3 good
**Rating:** 6
**Confidence:** 4

**Summary:**

The authors present, Cleanba, a new platform for distributed DRL that addresses reproducibility issues under different hardware settings. One of the biggest challenges in applying deep reinforcement learning to real world problems is reproducibility. In general, the algorithms are not robust to implementation details which are often crucial for successful real-world applications. The authors introduce a new distributed architecture that is reproducible if random seeds are controlled and leverages interleaving the actor and learner’s computations. The main contribution of the paper is that the authors propose a new mechanism to synchronize the actor and learner which reduces the non-determinism and improves reproducibility. While the synchronization mechanism is simple, the experiments show that Cleanba’s variants can obtain equivalent or higher scores than moolib’s IMPALA, but with 1) less training wall time under the 8 GPU setting and 2) more reproducible learning curves in different hardware settings.

**Strengths:**

Originality & Significance. The proposed synchronization mechanism is simple yet results in significant improvement in training time and reproducibility. Both reduced training time and reproducibility are among the key challenges in applying deep reinforcement learning to real works problems. Therefore despite the simplicity of the proposed mechanism, I believe that the performance improvement is significant.

Quality & Clarity. The authors describe the current state-of-the-art and its pitfalls clearly using clear code samples. They provide empirical results to further their hypothesis on what contribute to the current reproducibility issues in popular distributed reinforcement learning architectures. The experiments are extensive in comparing the proposed architecture to other benchmarks in the literature.

**Weaknesses:**

Originality & Significance. The authors propose to mechanisms to overcome the reproducibility issues in distributed reinforcement learning: 1) de-coupling hyper-parameters, 2) a new synchronization mechanism to better align actors and learners. The de-coupling mechanism is not novel and already exists in prior literature. The synchronization mechanism is simple yet novel. The performance of the new approach is tested only in Atari which is not a real-world problem where reproducibility is a key challenge. The paper can greatly benefit from a real-world application.


Quality & Clarity. The language is a bit off in several sections, specifically in the first paragraph, which makes it difficult to read at times. There are several grammatical errors as well.

**Questions:**

At first, I was really confused about the novelty and contribution of the paper. In the abstract and introduction, the paper talks about variants of already existing algorithms but does not talk about the novelty of the proposed variant specifically until past mid-way though the paper. It will be better to talk about the contributions first even when they may be simple yet have far reaching implications to reduce training time and improve reproducibility at scale.

It is important to consider a real-world example when reproducibility is one of the main challenges. Atari is a great way to benchmark but does not present all the variability such as in robotics applications for example to truly highlight issues related to the reproducibility.

In Section 5.1, the authors state that Cleanba’s IMPALA is 6.8x faster than monobeast’s IMPALA, mostly because Cleanba actors
run on GPUs, whereas monobeast’s actors run on CPUs. I think the performance metrics should be based on the same compute hardware CPU only or GPU only.

---

> ### Author Response · Authors · 2023-11-16
> **Rebuttal**
>
> We thank the reviewer’s helpful comments. We are glad they found our synchronization mechanism results in significant improvements in training time and reproducibility.
>
> > Q: The paper can greatly benefit from a real-world application. Atari is a great way to benchmark but does not present all the variability such as in robotics applications for example to truly highlight issues related to the reproducibility.
>
> The reproducibility issues related to working with physical robots seem to be in a completely different realm of reproducibility issues dealt with in this paper. Our work focuses on reproducibility issues related to scaling caused by distributed data structures and implementation.
>
> > Q: The language is a bit off in several sections, specifically in the first paragraph, which makes it difficult to read at times. There are several grammatical errors as well.
>
> We thank this constructive feedback and will get the paper proofread by a native speaker should the paper get accepted.
>
> > Q: In Section 5.1, the authors state that Cleanba’s IMPALA is 6.8x faster than monobeast’s IMPALA, mostly because Cleanba actors run on GPUs, whereas monobeast’s actors run on CPUs. I think the performance metrics should be based on the same compute hardware CPU only or GPU only.
>
> This is a great observation! GPU only is difficult because both monobeast’s IMPALA and the original IMPALA does not support GPU actors by default. Moolib however is a more fair comparison though, since it runs the actor on GPUs.

---

> > ### Comment · Reviewer_eKtC · 2023-11-22
> >
> > I would like to thank the authors for their response to my questions. My rating stays the same.

---

### Official Review · Reviewer_Ui8X · 2023-11-01

**Soundness:** 3 good
**Presentation:** 3 good
**Contribution:** 2 fair
**Rating:** 6
**Confidence:** 3

**Summary:**

This paper presents cleanba, a jax based software framework for reproducible distributed deep reinforcement learning (DDRL). The authors present errors that arise from existing non-deterministic DDRL frameworks that could harm reproducibility efforts, then propose a deterministic solution (which comes at the cost of stale data). The authors show the stability and reproducibility of cleanba, and evaluate it on a suite of standard Atari benchmarks.

**Strengths:**

- Reproducibility is a notorious issue in all of machine learning, but RL especially, and it is good to see more work addressing it
- The paper is generally clear and understandable
- Figure/text colouring is generally pretty useful

**Weaknesses:**

- The abstract’s first sentences feel a bit choppy
- There are other Sebulba implementations that might be worth comparing to (e.g. https://github.com/instadeepai/sebulba)
- Fig 2 could motivate the problem better, given that in the current figure both versions end up at very similar scores. If there was a different environment just highlighting that this 1 second lag has a meaningful detriment to final performance that could be really compelling.
- Although IMPALA is a common algorithm, devoting a short paragraph to it in the background could be helpful (since although many are familiar with it, not as many are with the intricacies as it relates to distributed computing that get mentioned in the next section)
- In Figure 5, the y axis could have their tickers removed. The absolute numeric scales are not particularly meaningful (I suspect even the most experienced RL researcher will not know if a score of 200,000 on UpNDown is good) and add a lot of clutter to the figure
- The majority of Fig 1 is highlighted. Maybe it’s better to just highlight the part that is the same? Or remove the highlighting? It just doesn’t add much when the points you want to draw attention to are basically the whole code block.
- In figure 3, the optimality gap x tickers are overlapping
- Moolib is a good point of comparison, but I see that this repo is now archived. Are there other DDRL libraries worth comparing to that are more actively maintained?
- One concern I have is that this paper tries too hard to be an algorithm paper. It presents cleanba, which is a software package that the authors want people to use, but not a lot of time is dedicated to that. More time is dedicated to the analysis of results from cleanba. While these results are somewhat interesting, I think they could be consolidated more (e.g. move the individual atari games to the appendix, and just present HNS, just an idea). Then use that space to actually describe the package of cleanba more (e.g. how does a user interact with it? Extend it? UML diagram perhaps? Etc.). The paper is called “cleanba” for the package, but I can’t say I really know programmatically much about cleanba after reading the paper.

**Questions:**

- I can’t evaluate the code, since there is no anonymized access to it, but I feel given that this introduces a package, that code quality is relevant. Is the code documented well and complete with type hints and docstrings and the like?
- How easy is it to extend the framework? What plans are there for other algorithms? What do the development goals look like from here?

---

> ### Author Response · Authors · 2023-11-16
> **Rebuttal**
>
> We thank the reviewer for the helpful feedback. We are glad the reviewer found the text/color helpful.
>
> > Q: The abstract’s first sentences feel a bit choppy
>
> We changed it to "Distributed Deep Reinforcement Learning (DRL) aims to train autonomous agents in less wall-clock time by leveraging more computational resources". This puts the parallel construction closer so less difficult to read.
>
> > Q: There are other Sebulba implementations that might be worth comparing to (e.g. https://github.com/instadeepai/sebulba)
>
> `instadeepai/sebulba` looks interesting and we are glad to see more researchers doing open source distributed DRL. Our work differs in several ways:
>
> * **instadeep/sebulba and cleanba gets similar end-to-end performance**, both using about 11 minutes to achieve 800 scores in Breakout when using 128 accelerator cores (see our Appendix F and Figure 5 at their [blog post](https://cloud.google.com/blog/products/compute/instadeep-performs-reinforcement-learning-on-cloud-tpus)).
> * instadeep/sebulba’s PPO has higher FPS but lower data efficiency: instadeep/sebulba’s PPO can reach a 2.25x higher FPS than cleanba’s PPO. Cleanba gets 1.6M FPS with 128 A100s, whereas instadeep/sebulba get 3.6M FPS with 128 TPUv4 cores, but since the end-to-end performance is similar, we can see that instadeep/sebulba’s PPO’s speed up is offset by the reduction in data efficiency.
> * instadeep/sebulba mainly focus on the FPS scaling and end-to-end experiments in Breakout, while our work is more comprehensive (e.g., 57 Atari games, more random seeds, support both IMPALA and PPO, comparison w/ moolib and torchbeast, reproducibility under different hardware settings)
>
> > Fig 2 could motivate the problem better, given that in the current figure both versions end up at very similar scores. If there was a different environment just highlighting that this 1 second lag has a meaningful detriment to the final performance could be really compelling.
>
> Thank you for this suggestion. Our cluster is under heavy usage at the moment but happy to explore this later (unfortunately after the discussion period). We may just run these experiments on Breakout since we expect to see a much larger difference.
>
> > In figure 3, the optimality gap x tickers are overlapping
>
> Thank you. This is fixed.
>
> > Moolib is a good point of comparison, but I see that this repo is now archived. Are there other DDRL libraries worth comparing to that are more actively maintained?
>
> There are some DDRL libraries that we thought about comparing but chose not to do it for the following reasons.
>
> 1. Ray’s RLlib: we are not aware of its distributed IMPALA reproducing the level of performance in Atari. For example, the following table shows RLlib’s official benchmark at https://github.com/ray-project/rl-experiments#impala-and-a2c ran for 4 (out of 57) games and significantly underperformed IMPALA Shallow under an hour of training
>
> | | RLlib IMPALA 32-workers | IMPALA Shallow (Table C.1, Espeholt et al 2018) |
> | --- | --- | --- |
> | BeamRider | 3,181 | 32,463.47 |
> | Breakout | 538 | 640.43 |
> | QBert | 10,850 | 351,200.12 |
> | Spacelnvaders | 843 | 43,595.78 |
>
> 2. DeepMind/acme: we noted that its technical report [V2](https://arxiv.org/pdf/2006.00979.pdf) contained only Atari evaluations on 5 (out of 57) games and revealed no runtime information. Its [v1](https://arxiv.org/pdf/2006.00979v1.pdf) revealed runtime information IMPALA but it seems quite different from the original IMPALA results
>   * For example, ACME’s IMPALA takes 12 hours to get 100k in Asteroids, 24 hours to get 750 in Breakout, but original IMPALA (deep) reported 100k in Asteroids and 783 in Breakout in ~2 hours (the runtime is interpolated based on Figure 2 in https://openreview.net/pdf?id=r1lyTjAqYX)
>   * We also tried running ACME’s IMPALA and got very low FPS (like 1200 FPS) both in single GPU mode and distributed node for some reason… We suspect a higher performance of ACME might be coupled with DeepMind’s internal infrastructure.
>
> We chose to evaluate against moolib and tourchbeast because they, to our knowledge, are the **only** open-source DDRL infrastructures that have evaluated their performance on 50+ Atari games using the 200M frames. 200M frames also make the computational resources reasonable for us (about 2 hours per game with a single GPU or 20-30 minutes per game with 8 GPUs), as opposed to Seed RL which is evaluated on 40B frames (about 40 hours using their settings — 8xTPUv3, 213 CPUs)
>
> The lack of highly reproducible and easy-to-understand DDRL infrastructure is what partly motivated this work — we aim to make DDRL more accessible and transparent for the research community.

---

> ### Author Response · Authors · 2023-11-16
> **Rebuttal (cont'd)**
>
> > Q: I can’t evaluate the code, since there is no anonymized access to it, [...] The paper is called “cleanba” for the package, but I can’t say I really know programmatically much about cleanba after reading the paper.
>
> We made the anonymized version of Cleanba available at https://anonymous.4open.science/r/cleanba-test-9868/README.md. It’s basically two files: cleanba_ppo.py and cleanba_impala.py, both of which are self-contained and within 800 lines of code.
>
> We just want to provide simple-to-understand DDRL infrastructure which is battle-tested and scalable in various scenarios, so we ran extensive evaluation on 57 Atari games and reproducibility under different hardware scenarios.
>
> On the other end of the spectrum, we have libraries like RLlib / ACME, which are monolithic. Have tons of abstraction and modules (ACME has 500 python files, and Ray (including) has 2.8k Python files). A motivating (dummy) question for us is “Do we really need to read hundreds of interconnected files to understand how DDRL works from end to end?”
>
> The main target audience is students and researchers, so they can learn DDRL with a principled approach and hack it for their respective research, similar to CleanRL’s approach to DRL infrastructure (https://www.jmlr.org/papers/v23/21-1342.html).
>
> > Is the code documented well and complete with type hints and docstrings and the like?
>
> Cleanba’s implementations are extremely lightweight: they have minimal lines of code and some necessary comments but come with extensive documentation, evaluation, and documentation in the repo/paper. We note that since each Cleanba variant is only 800 lines of code end-to-end without extra modules, adding extensive docstring may not be necessary.
>
> The users can easily trace the flow of the program all happening in a single file (e.g., cleanba_ppo.py), as opposed to DRL libraries like SB3, where it flows through dozens of files like below. In SB3, since there are hundreds of classes and methods involved, docstrings are necessary. In cleanba_ppo.py, there are only 7 classes and 16 functions, all of which are self-explanatory.
>
> The list of involved files in SB3 (taken from Appendix B in https://www.jmlr.org/papers/volume23/21-1342/21-1342.pdf)
> 1. stable baselines3/ppo/ppo.py — 315 lines of code (LOC), 51 lines of docstring (LOD)
> 2. stable baselines3/common/on policy algorithm.py — 280 LOC, 49 LOD
> 3. stable baselines3/common/base class.py — 819 LOC, 231 LOD
> 4. stable baselines3/common/utils.py — 506 LOC, 195 LOD
> 5. stable baselines3/common/env util.py — 157 LOC, 43 LOD
> 6. stable baselines3/common/atari wrappers.py — 249 LOC, 84 LOD
> 7. stable baselines3/common/vec env/ init .py — 73 LOC, 24 LOD
> 8. stable baselines3/common/vec env/dummy vec env.py — 126 LOC, 25 LOD
> 9. stable baselines3/common/vec env/base vec env.py — 375 LOC, 112 LOD
> 10. stable baselines3/common/vec env/util.py — 77 LOC, 31 LOD
> 11. stable baselines3/common/vec env/vec frame stack.py — 65 LOC, 14 LOD
> 12. stable baselines3/common/vec env/stacked observations.py — 267 LOC, 74 LOD
> 13. stable baselines3/common/preprocessing.py — 217 LOC, 68 LOD
> 14. stable baselines3/common/buffers.py — 770 LOC, 183 LOD
> 15. stable baselines3/common/policies.py — 962 LOC, 336 LOD
> 16. stable baselines3/common/torch layers.py — 318 LOC, 97 LOD
> 17. stable baselines3/common/distributions.py — 700 LOC, 228 LOD
> 18. stable baselines3/common/monitor.py — 240 LOC, 76 LOD
> 19. stable baselines3/common/logger.py — 640 LOC, 201 LOD
> 20. stable baselines3/common/callbacks.py — 603 LOC, 150 LOD
>
> > How easy is it to extend the framework? What plans are there for other algorithms? What do the development goals look like from here?
>
> Extending Cleanba is pretty straightforward: the researcher can just clone the Cleanba variants and make whatever modifications they desire. Cleanba variants have less than 800 lines of code and make minimal abstractions/modularities, so it opens up a lot of possibilities for developing customized features.
>
> We do not plan to add other algorithms at this time. The purpose of Cleanba may be similar to https://github.com/karpathy/nanoGPT: small, clean, and highly hackable, and has strong empirical performance.

---

> > ### Comment · Reviewer_Ui8X · 2023-11-19
> >
> > I appreciate the authors responses, and they have addressed many of my concerns. As such I am upgrading my score (5->6)

---

### Official Review · Reviewer_Z35M · 2023-11-01

**Soundness:** 2 fair
**Presentation:** 2 fair
**Contribution:** 2 fair
**Rating:** 3
**Confidence:** 4

**Summary:**

This paper introduces Cleanba, a new reproducible distributed RL training platform. It verifies the reproducibility on PPO and IMPALA in different hardware settings.

**Strengths:**

It implements a new reproducible and fast distributed RL training platform.

**Weaknesses:**

1) The paper does not touch the real problems that cause the reproducibility issues. It not only occurs in distributed RL training but also occurs in the single machine training even using the same hyperparameters. Reverb and [1] have reported that replay ratio (the number of gradient updates per environment transition) greatly affects the performance. [1] Revisiting Fundamentals of Experience Replay
2) The Cleanba shown in Figure 1 does not match the “learner always learns from the rollout data obtained from the second latest policy” (line 7).
3) “ensuring the learner performs gradient updates with rollout data of second latest policy” assumption is too strong. It may slow down the IMPALA training.
4) It would be beneficial to provide more reproducibility results in relation to different RL algorithms such as Ape-X, SeedRL, etc.

**Questions:**

Please address the above weaknesses.

---

> ### Author Response · Authors · 2023-11-16
> **Rebuttal**
>
> > Q: The paper does not touch the real problems that cause the reproducibility issues. It not only occurs in distributed RL training but also occurs in the single machine training even using the same hyperparameters.
>
> Yes, we absolutely agree that reproducibility issues can occur in single-machine settings even using the same hyperparameters since we are dealing with stochastic environments. Some environments can be more brittle than others. For example, in SpaceInvaders-v5, two of the Cleanba PPO (Sync) runs get ~20,000 scores, but one run got ~2,000 scores, whereas all three runs of Cleanba PPO (Sync) get ~450 scores.
> In this paper, we are focusing on reproducibility issues related to scaling caused by distributed data structures and implementation.
>
> >Q: Reverb and [1] have reported that replay ratio (the number of gradient updates per environment transition) greatly affects the performance. [1] Revisiting Fundamentals of Experience Replay
>
> Thanks for referencing this very interesting work on *off-policy* DRL algorithms, however, it may not be related to our work which focuses on *on-policy* algorithms like PPO and IMPALA. In particular, PPO and IMPALA do not have a replay buffer like in reverb; if we tried to use [1]’s terminology, then the *age of a transition* is always 1, and the *age of the oldest policy* is always 2 under the Cleanba architecture.
>
> > Q: The Cleanba shown in Figure 1 does not match the “learner always learns from the rollout data obtained from the second latest policy” (line 7).
>
> There is a pre-condition “**starting from iteration 3** the learner always learns from the rollout data obtained from the second latest policy”; before iteration 3, the learner does not necessarily learn from the second latest policy.
>
> > Q: “ensuring the learner performs gradient updates with rollout data of second latest policy” assumption is too strong. It may slow down the IMPALA training.
>
> Though this may be a possibility, our experiments show otherwise: Cleanba IMPALA has obtained equivalent scores and faster training time than monobeast and moolib’s IMPALA. Cleanba’s IMPALA is faster than monobeast’s IMPALA because Cleanba IMPALA run actors on GPU while monobeast’s IMPALA run actors on CPU. Cleanba’s IMPALA is faster than moolib’s IMPALA likely because Cleanba’s JAX + EnvPool is faster than Moolib’s PyTorch + various C++ async operations (e.g., async accumulation of gradients, async environment stepping, async network evaluation).
>
> > Q: It would be beneficial to provide more reproducibility results in relation to different RL algorithms such as Ape-X, SeedRL, etc.
>
> To the best of our knowledge, there is no open-source Ape-X replication that has public results showing that they could reproduce 350% median HNS in 20 hours. Without reliable Ape-X infrastructure, it’s not viable to provide reproducibility results.
>
> Reproducibility results with Seed RL is challenging for several reasons:
> * Seed RL is already unmaintained and running its code can be a challenge.
>   * We tried to follow its setup instructions “./run_local.sh atari r2d2 4 4" and it immediately errored out with some GPG errors `W: GPG error: https://developer.download.nvidia.com/compute/cuda/repos/ubuntu1804/x86_64  InRelease: The following  signatures couldn't be verified because the public key is not available: NO_PUBKEY A4B469963BF863CC`
>   * We then tried fixing it with https://github.com/NVIDIA/nvidia-docker/issues/1632#issuecomment-1112667716 and got
>       ```ERROR: Could not find a version that satisfies the requirement ale-py~=0.8.0; extra == "atari" (from gym[atari]) (from versions: 0.6.0.dev20200207, 0.7rc0, 0.7rc1, 0.7rc2, 0.7rc3, 0.7rc4, 0.7, 0.7.1)```
>   * We then kept trying to fix it for 30 minutes to no avail and it was just an error every step of the way (e.g., cmake errors)
>
> Its repository contains many reproducibility-related issues:
> * unable to reproduce FPS (https://github.com/google-research/seed_rl/issues/80)
> * unable to reproduce Pong results (https://github.com/google-research/seed_rl/issues/75, https://github.com/google-research/seed_rl/issues/51)
> * does not work as expected with TPU / never tested end-to-end results with GPUs (https://github.com/google-research/seed_rl/issues/78#issuecomment-1258515133)
> * error with multiple GPUs (https://github.com/google-research/seed_rl/issues/16)
>
> We chose to evaluate against moolib and tourchbeast because they, to our knowledge, are the only open-source DDRL infrastructures which have evaluated their performance on 50+ Atari games using the 200M frames. 200M frames also make the computational resources reasonable for us (about 2 hours per game with a single GPU or 20-30 minutes per game with 8 GPUs), as opposed to Seed RL which is evaluated on 40B frames (about 40 hours using their settings — 8xTPUv3, 213 CPUs)

---

> > ### Comment · Reviewer_Z35M · 2023-11-22
> >
> > Thank you for the author's response. It addresses some of my questions. However, my main concern regarding the generalization of the proposed methods to different types of RL algorithms remains unaddressed. Additionally, learning from rollout data of second latest policy cannot provide a mathematical guarantee for exact result reproduction of original on-policy RL algorithms.

---

> > > ### Author Response · Authors · 2023-11-22
> > > **Reply**
> > >
> > > Thank you for the response. We are glad it addresses some of your questions.
> > >
> > > > Additionally, learning from rollout data of second latest policy cannot provide a mathematical guarantee for exact result reproduction of original on-policy RL algorithms.
> > >
> > > If we control the environment data generation and GPU non-determinism, it is possible to get **exact results**:
> > >
> > > Here is the cleanba's architecture using `torch.nn.Linear(1,1)`, which gets identical trained model parameters across runs.
> > > ```
> > > import queue
> > > import threading
> > > import torch
> > > batch_size = 32
> > > agent = torch.nn.Linear(1, 1)
> > > optim = torch.optim.SGD(agent.parameters(), lr=0.1)
> > > torch.nn.init.constant_(agent.weight, 0.5)
> > > torch.nn.init.constant_(agent.bias, 0.5)
> > > rollout_Q = queue.Queue(maxsize=1)
> > > param_Q = queue.Queue(maxsize=1)
> > > ITER = 100
> > > def actor():
> > >     actor_agent = torch.nn.Linear(1, 1) # dummy initialization
> > >     for i in range(1, ITER):
> > >         if i != 2:
> > >             params = param_Q.get()
> > >             actor_agent.load_state_dict(params)
> > >         # dummy data `rollout(params, batch_size)`
> > >         data = torch.ones(batch_size, 1) + i + actor_agent.weight
> > >         rollout_Q.put(data)
> > > def learner():
> > >     for i in range(1, ITER-1):
> > >         data = rollout_Q.get()
> > >         if i == 1:
> > >             print(f"{data.flatten()=}, {data.shape=}")
> > >         # agent.learn(data)
> > >         loss = agent(data).mean()
> > >         loss.backward()
> > >         optim.step()
> > >         param_Q.put(agent.state_dict())
> > > param_Q.put(agent.state_dict())
> > > threads = [threading.Thread(target=actor), threading.Thread(target=learner)]
> > > for thread in threads:
> > >     thread.start()
> > > for thread in threads:
> > >     thread.join()
> > > params = param_Q.get()
> > > trained_agent = torch.nn.Linear(1, 1)  # dummy initialization
> > > trained_agent.load_state_dict(params)
> > > print(trained_agent.weight, trained_agent.bias)
> > > ```
> > > ```
> > > (cleanba-py3.9) ➜  cleanba-test git:(main) ✗ python  cleanba_pseudocode.py
> > > data.flatten()=tensor([2.5000, 2.5000, 2.5000, 2.5000, 2.5000, 2.5000, 2.5000, 2.5000, 2.5000,
> > >         2.5000, 2.5000, 2.5000, 2.5000, 2.5000, 2.5000, 2.5000, 2.5000, 2.5000,
> > >         2.5000, 2.5000, 2.5000, 2.5000, 2.5000, 2.5000, 2.5000, 2.5000, 2.5000,
> > >         2.5000, 2.5000, 2.5000, 2.5000, 2.5000], grad_fn=<ViewBackward0>), data.shape=torch.Size([32, 1])
> > > Parameter containing:
> > > tensor([[-287.2436]], requires_grad=True) Parameter containing:
> > > tensor([-484.6000], requires_grad=True)
> > > (cleanba-py3.9) ➜  cleanba-test git:(main) ✗ python  cleanba_pseudocode.py
> > > data.flatten()=tensor([2.5000, 2.5000, 2.5000, 2.5000, 2.5000, 2.5000, 2.5000, 2.5000, 2.5000,
> > >         2.5000, 2.5000, 2.5000, 2.5000, 2.5000, 2.5000, 2.5000, 2.5000, 2.5000,
> > >         2.5000, 2.5000, 2.5000, 2.5000, 2.5000, 2.5000, 2.5000, 2.5000, 2.5000,
> > >         2.5000, 2.5000, 2.5000, 2.5000, 2.5000], grad_fn=<ViewBackward0>), data.shape=torch.Size([32, 1])
> > > Parameter containing:
> > > tensor([[-287.2436]], requires_grad=True) Parameter containing:
> > > tensor([-484.6000], requires_grad=True)
> > > (cleanba-py3.9) ➜  cleanba-test git:(main) ✗ python  cleanba_pseudocode.py
> > > data.flatten()=tensor([2.5000, 2.5000, 2.5000, 2.5000, 2.5000, 2.5000, 2.5000, 2.5000, 2.5000,
> > >         2.5000, 2.5000, 2.5000, 2.5000, 2.5000, 2.5000, 2.5000, 2.5000, 2.5000,
> > >         2.5000, 2.5000, 2.5000, 2.5000, 2.5000, 2.5000, 2.5000, 2.5000, 2.5000,
> > >         2.5000, 2.5000, 2.5000, 2.5000, 2.5000], grad_fn=<ViewBackward0>), data.shape=torch.Size([32, 1])
> > > Parameter containing:
> > > tensor([[-287.2436]], requires_grad=True) Parameter containing:
> > > tensor([-484.6000], requires_grad=True)
> > > ```

---

> > > > ### Author Response · Authors · 2023-11-22
> > > > **Reply (cont'd)**
> > > >
> > > > Here is IMPALA's code under similar settings. Note that even the first data even becomes different due to the non-determinism arising from the thread execution. Unsurprisingly, the final model parameters are not reproducible under this example.
> > > >
> > > > The `broadcast_to_actors(agent.param)` logic is implemented as `global_actor_agent.load_state_dict(agent.state_dict())`, which is taken directly from torchbeast's IMPALA implementation (https://github.com/facebookresearch/torchbeast/blob/0af07b051a2176a8f9fd20c36891ba2bba6bae68/torchbeast/monobeast.py#L295)
> > > >
> > > > ```
> > > > import queue
> > > > import threading
> > > > import torch
> > > > batch_size = 32
> > > > agent = torch.nn.Linear(1, 1)
> > > > global_actor_agent = torch.nn.Linear(1, 1)
> > > > optim = torch.optim.SGD(agent.parameters(), lr=0.1)
> > > > torch.nn.init.constant_(agent.weight, 0.5)
> > > > torch.nn.init.constant_(agent.bias, 0.5)
> > > > rollout_Q = queue.Queue()
> > > > terminate_Q = queue.Queue()
> > > > ITER = 100
> > > > def actor():
> > > >     i = 0
> > > >     while True:
> > > >         # dummy data `rollout(params, 1)`
> > > >         data = torch.ones(1) + i + global_actor_agent.weight
> > > >         try:
> > > >             signal = terminate_Q.get(timeout=0.0001)
> > > >             if signal == 'terminate':
> > > >                 break
> > > >         except queue.Empty:
> > > >             pass
> > > >         rollout_Q.put(data)
> > > >         i += 1
> > > > def learner():
> > > >     for i in range(1, ITER-1):
> > > >         data = torch.stack([rollout_Q.get()[0] for _ in range(batch_size)])
> > > >         if i == 1:
> > > >             print(f"{data.flatten()=}, {data.shape=}")
> > > >         # agent.learn(data)
> > > >         loss = agent(data).mean()
> > > >         loss.backward()
> > > >         optim.step()
> > > >         # https://github.com/facebookresearch/torchbeast/blob/0af07b051a2176a8f9fd20c36891ba2bba6bae68/torchbeast/monobeast.py#L295
> > > >         global_actor_agent.load_state_dict(agent.state_dict())
> > > > learner_thread = threading.Thread(target=learner)
> > > > actor_threads = [threading.Thread(target=actor) for _ in range(batch_size)]
> > > > learner_thread.start()
> > > > for thread in actor_threads:
> > > >     thread.start()
> > > > learner_thread.join()
> > > > for thread in actor_threads:
> > > >     terminate_Q.put('terminate')
> > > > for thread in actor_threads:
> > > >     thread.join()
> > > > print(agent.weight, agent.bias)
> > > > ```
> > > > ```
> > > > (cleanba-py3.9) ➜  cleanba-test git:(main) ✗ python impala_pseudocode.py
> > > > data.flatten()=tensor([1.8847, 1.8847, 1.8847, 1.8847, 1.8847, 2.8847, 2.8847, 2.8847, 1.8847,
> > > >         2.8847, 2.8847, 1.8847, 3.8847, 3.8847, 3.8847, 3.8847, 3.8847, 2.8847,
> > > >         4.8847, 1.8847, 2.8847, 4.8847, 1.8847, 4.8847, 4.8847, 5.8847, 1.8847,
> > > >         3.8847, 2.8847, 2.8847, 3.8847, 1.8847], grad_fn=<ViewBackward0>), data.shape=torch.Size([32, 1])
> > > > Parameter containing:
> > > > tensor([[-16444.9629]], requires_grad=True) Parameter containing:
> > > > tensor([-484.6000], requires_grad=True)
> > > > (cleanba-py3.9) ➜  cleanba-test git:(main) ✗ python impala_pseudocode.py
> > > > data.flatten()=tensor([1.2628, 1.2628, 1.2628, 1.2628, 1.2628, 2.2628, 2.2628, 1.2628, 2.2628,
> > > >         1.2628, 1.2628, 1.2628, 2.2628, 3.2628, 3.2628, 4.2628, 2.2628, 1.2628,
> > > >         2.2628, 5.2628, 2.2628, 4.2628, 2.2628, 2.2628, 2.2628, 3.2628, 3.2628,
> > > >         3.2628, 3.2628, 3.2628, 6.2628, 1.2628], grad_fn=<ViewBackward0>), data.shape=torch.Size([32, 1])
> > > > Parameter containing:
> > > > tensor([[-16193.5635]], requires_grad=True) Parameter containing:
> > > > tensor([-484.6000], requires_grad=True)
> > > > (cleanba-py3.9) ➜  cleanba-test git:(main) ✗ python impala_pseudocode.py
> > > > data.flatten()=tensor([0.7266, 0.7266, 0.7266, 0.7266, 0.7266, 1.7266, 0.7266, 1.7266, 1.7266,
> > > >         1.7266, 1.7266, 2.7266, 0.7266, 1.7266, 0.7266, 2.7266, 0.7266, 3.7266,
> > > >         0.7266, 2.7266, 1.7266, 4.7266, 2.7266, 2.7266, 0.7266, 2.7266, 1.7266,
> > > >         1.7266, 2.7266, 3.7266, 1.7266, 0.7266], grad_fn=<ViewBackward0>), data.shape=torch.Size([32, 1])
> > > > Parameter containing:
> > > > tensor([[-16209.0605]], requires_grad=True) Parameter containing:
> > > > tensor([-484.6000], requires_grad=True)
> > > > ```

---

> > > > > ### Author Response · Authors · 2023-11-22
> > > > > **Reply (cont'd)**
> > > > >
> > > > > The reason that the cleanba PPO or IMPALA can get different learning curves is primarily due to GPU non-determinism. As Agarwal et al (2021) pointed out, "A run can be different from using a fixed random seed. Indeed, fixing the seed may not be able to control all sources of randomness, such as non-determinism of ML frameworks with GPUs" (also see their Figure A.13)
> > > > >
> > > > > In this work, we at least make sure the cleanba architecture itself is deterministic, so if we see unreproducible results, we know they come from other sources of non-determinism (e.g., environment, neural network initialization) instead of coming from the DDRL architecture. In IMPALA, unreproducible results could come from the non-determinism arising from concurrent thread scheduling, as shown in our paper and the toy example shown in the previous reply.
> > > > >
> > > > >
> > > > > Agarwal, R., Schwarzer, M., Castro, P. S., Courville, A. C., & Bellemare, M. (2021). Deep reinforcement learning at the edge of the statistical precipice. Advances in neural information processing systems, 34, 29304-29320. https://arxiv.org/pdf/2108.13264.pdf

---

> ### Comment · Reviewer_Z35M · 2023-11-23
>
> Thank you for the author's response. My question is that Cleanba cannot exact reproduce the original PPO algorithm althrough Cleanba can reproduce itself without environment randomness and GPU non-determinism.

---

### Official Review · Reviewer_fstX · 2023-11-07

**Soundness:** 3 good
**Presentation:** 3 good
**Contribution:** 3 good
**Rating:** 6
**Confidence:** 3

**Summary:**

The authors introduce CleanBa, a distributed RL library containing distributed implementations of IMPALA and PPO. They demonstrate that IMPALA's actor-learner architecture causes issues with reproducibility
because the learner can do a different number of updates depending on its speed.

To solve this, they then propose a new architecture, where the policy updates based on the rollouts of the second latest policy. This ensures reproducibility at the cost of more synchronisation and the introduction of stale data.

They demonstrate that their framework performs well compared to relevant baselines and runs much more quickly.

**Strengths:**

* The paper is well written and clearly explained.
* The reproducibility problems identified in IMPALA are interesting and well addressed by their new framework.
* The empirical results demonstrate the framework is faster than prior methods on comparable hardware.

**Weaknesses:**

* The authors claim to have built a distributed learning framework, but have only evaluated their method on a single machine. This is a major flaw in their evaluations, and the reason I recommend rejection for this paper. If this is included I will increase my score.


Typos:
At the end of section 4.2 you write `jax.distibuted` -- I think this should be `jax.distributed` (extra r).

**Questions:**

* How does the extra param queue impact the distributed performance (i.e. speed)? i.e. how much cost is paid in speed for fixing the reproducibility issues?

---

> ### Author Response · Authors · 2023-11-16
> **Rebuttal**
>
> We thank the reviewer for the helpful feedback, and we are glad they found the reproducibility problem identified in IMPALA interesting.
>
>
> > Q: The authors claim to have built a distributed learning framework, but have only evaluated their method on a single machine. This is a major flaw in their evaluations, and the reason I recommend rejection for this paper. If this is included I will increase my score.
>
> We thank the reviewer for this constructive feedback. Cleanba **can** scale to multi-node and we have included evaluations of multi-node training in Appendix F. Multi-node training is computationally expensive, so we focus evaluation specifically on a representative Atari game Breakout instead of the full Atari suite. We have also mentioned the multi-node scaling results briefly in the main text.
>
>
> > Q: Typos: At the end of section 4.2 you write jax.distibuted -- I think this should be jax.distributed (extra r).
>
> Thanks for the correction on our typo.
>
> > Q: How does the extra param queue impact the distributed performance (i.e., speed)? i.e. how much cost is paid in speed for fixing the reproducibility issues?
>
> There are some idle times associated with ensuring the actor always learns from the second latest policy. We measure the estimated training time and rollout time, estimate how much percentage of the actor and learner computations can overlap (`min(training time, rollout time) / max(training time, rollout time)`), and as a result, how much time either the learner or actor is idling `( max(training time, rollout time) - min(training time, rollout time)) * the number of updates`.
>
> |                                    | training time   | rollout time    | learner / actor computation overlap percentage | actor / learner idle time (mins) |
> |------------------------------------|-----------------|-----------------|------------------------------------------------|------------------------------------|
> | Cleanba IMPALA, 1 A100, 10 CPU     | 0.1130 ± 0.0003 | 0.3864 ± 0.0045 | 29.24%                                         | 88.99625667                        |
> | Cleanba IMPALA, 8 A100, 46 CPU     | 0.0453 ± 0.0004 | 0.0585 ± 0.0002 | 77.44%                                         | 4.29682                            |
> | Cleanba PPO (Sync), 1 A100, 10 CPU | 2.0839 ± 0.0024 | 2.2496 ± 0.0264 | -                                              | -                                  |
> | Cleanba PPO (Sync), 8 A100, 46 CPU | 0.3257 ± 0.0004 | 0.4945 ± 0.0010 | -                                              | -                                  |
> | Cleanba PPO, 8 A100, 46 CPU        | 0.5720 ± 0.0058 | 0.4198 ± 0.0016 | 73.39%                                         | 7.73937                            |
> | Cleanba PPO, 1 A100, 10 CPU        | 2.1270 ± 0.0005 | 3.6205 ± 0.3244 | 58.75%                                         | 75.944475                          |
>
> Take the second row as an example, the training time and rollout time of Cleanba IMPALA, 8 A100, 46 CPU is roughly similar, so they can overlap 77.44% of learner and actor’s computation. We then get 0.0585 * 19531 = 1142 seconds ~= 19 minutes of the estimated total runtime (roughly similar to the empirical total runtime of 22 minutes which included initial compilation time and other overhead), during which the learner idled for (0.0585 - 0.0453) * 19531 ~= 257 seconds ~= 4 minutes
>
> While the idle time may seem like a lot, it’s unclear whether removing these idle times can result in faster end-to-end training. Without the restriction that the agent can only learn from the second latest policy, the agent may learn from more stale data. Take the recent result from instadeep/sebulba (https://cloud.google.com/blog/products/compute/instadeep-performs-reinforcement-learning-on-cloud-tpus), which removes the aforementioned restriction. As a result, their PPO runs seemingly faster and get ~2.2x higher FPS than Cleanba’s PPO. However, the end-to-end training results are almost identical — both achieved about ~800 scores in Breakout-v5 in 10 minutes.

---

> > ### Author Response · Authors · 2023-11-22
> > **Further comments on distributed evaluation**
> >
> > > The authors claim to have built a distributed learning framework, but have only evaluated their method on a single machine.
> >
> > Our core contribution is identifying and resolving reproducibility issues arising from distributed setups, so we obtain good reproducibility principles that help us scale to hundreds of GPUs. The definition of distributed setups also is not necessarily limited to multi-node settings. For example, in IMPALA [1] or Apex-DQN [2] "distributed" could mean the actor's computation is distributed to multiple CPUs instead of a single CPU or just refer to the actor-learner architecture.
> >
> > Regarding the main experiments of 57 Atari games, evaluating DDRL codebases on a single machine setting is a standard practice among prior DDRL works, such as Seed RL [3], IMPALA, Apex-DQN. In particular, IMPALA and Apex-DQN's main experiments use a single GPU in a single machine, whereas Seed RL's main experiments use a TPU v3, 8 cores with 213 cores machine.
> >
> > It is only when exploring the FPS settings did these DDRL frameworks evaluated the performance in a multi-node setting (such as Seed RL's Table 1, which evaluated only FPS and no human normalized score across 57 Atari games).
> >
> > To this end, we also included multi-node experiments scaling up to 128 GPUs in Appendix F. In these multi-node experiments we mostly care about walltime learning speed, so we do not need to worry about sample efficiency. We have simply kept doubling the simulation environments and CPUs to obtain higher FPS.
> >
> > * **Linear scaling w/ 97% of ideal scaling efficiency**. In 16 GPU, the FPS is 258754, and in 64 GPU, the FPS is 1013163, which achieves 1013163/(258754 * 4) = 97% of the ideal scaling efficiency. This is likely empowered by the fast connectivity offered by NVIDIA GPUDirect RDMA (remote direct memory access) in our cluster.
> > * **Small batch sizes train more efficiently**. As we increase batch sizes, particularly in the
> > first 40M steps, the sample efficiency tends to decline. This outcome is unsurprising, given
> > that the initial policy is random and Breakout initially has limited explorable game states.
> > In this case, the data in the batch is going to have less diverse data, which makes the large
> > batch size less valuable.
> > * **Large batch sizes train more quickly**. Like [4], we find increasing
> > the batch size does make the agent reach some given scores faster. This suggests that we
> > could always increase the batch size to obtain shorter training times if sample efficiency is
> > not a concern.
> >
> >
> > References:
> >
> > * [1] Lasse Espeholt, Hubert Soyer, Remi Munos, Karen Simonyan, Volodymyr Mnih, Tom Ward, Yotam ´
> > Doron, Vlad Firoiu, Tim Harley, Iain Dunning, Shane Legg, and Koray Kavukcuoglu. IMPALA: scalable distributed deep-rl with importance weighted actor-learner architectures. In Jennifer G. Dy and Andreas Krause (eds.), Proceedings of the 35th International Conference on
> > Machine Learning, ICML 2018, Stockholmsmassan, Stockholm, Sweden, July 10-15, 2018 ¨ , volume 80 of Proceedings of Machine Learning Research, pp. 1406–1415. PMLR, 2018. URL
> > http://proceedings.mlr.press/v80/espeholt18a.html
> > * [2] Dan Horgan, John Quan, David Budden, Gabriel Barth-Maron, Matteo Hessel, Hado van Hasselt,
> > and David Silver. Distributed prioritized experience replay. In 6th International Conference
> > on Learning Representations, ICLR 2018, Vancouver, BC, Canada, April 30 - May 3, 2018,
> > Conference Track Proceedings. OpenReview.net, 2018. URL https://openreview.net/
> > forum?id=H1Dy---0Z.
> > * [3] Lasse Espeholt, Raphael Marinier, Piotr Stanczyk, Ke Wang, and Marcin Michalski. Seed rl: ¨
> > Scalable and efficient deep-rl with accelerated central inference. In International Conference on Learning Representations, 2020. URL https://openreview.net/forum?id=
> > rkgvXlrKwH.
> > * [4] Sam McCandlish, Jared Kaplan, Dario Amodei, and OpenAI Dota Team. An empirical model of
> > large-batch training. arXiv preprint arXiv:1812.06162, 2018.

---

> > > ### Comment · Reviewer_fstX · 2023-11-22
> > >
> > > Thanks very much for your thorough reply to my comment -- it was interesting to read.
> > >
> > > > Cleanba can scale to multi-node and we have included evaluations of multi-node training in Appendix F
> > >
> > > My apologies! I completely missed this section. I am usually careful to check appendices for results I think are missing but obviously somehow missed them on this occasion.
> > >
> > > I will update my score to vote for acceptance.

---

### Author Response · Authors · 2023-11-16
**General comment on source code**

We thank the reviewers for their helpful feedback and comments. We have made the anonymized source code available at https://anonymous.4open.science/r/cleanba-test-9868/README.md for the reviewer's evaluation.

---

### Meta-Review · Area_Chair_Tkwe · 2023-12-07

**Metareview:**

This paper introduces a new framework for running a distributed version of IMPALA and PPO.

The strengths of the paper are an open-sourced, well tested and scalable approach to distributed RL that outperformes current methods based on the empirical analysis across a range of different hardware budgets.

A weakness is the lack of any theoretical guarantees and the narrow set of evaluation tasks (only Atari).

However, overall I believe that this is a thorough piece of work and will help the community develop better scalable RL algorithms. I thus recommend acceptance of the paper.
For the camera ready copy I recommend that the authors evaluate their method on other RL environments, e.g. challenging continuous control tasks. It would also be good to compare and contrast to recent end-to-end JAX based approaches since they offer an orthogonal axis towards scalability (without being distributed).

**Justification For Why Not Higher Score:**

See "weakness" above

**Justification For Why Not Lower Score:**

I believe this is a meaningful contribution to the community.

---

### Decision · Program_Chairs · 2024-01-16

Accept (poster)